# Molecular basis of signaling specificity between GIRK channels and GPCRs

Kouki K Touhara, Roderick MacKinnon*

Laboratory of Molecular Neurobiology and Biophysics, Howard Hughes Medical Institute, The Rockefeller University, New York, United States

**Abstract** Stimulated muscarinic acetylcholine receptors (M2Rs) release $G\beta\gamma$ subunits, which slow heart rate by activating a G protein-gated $K^+$ channel (GIRK). Stimulated $\beta2$ adrenergic receptors ($\beta2ARs$) also release $G\beta\gamma$ subunits, but GIRK is not activated. This study addresses the mechanism underlying this specificity of GIRK activation by M2Rs. $K^+$ currents and bioluminescence resonance energy transfer between labelled G proteins and GIRK show that M2Rs catalyze $G\beta\gamma$ subunit release at higher rates than $\beta2ARs$, generating higher $G\beta\gamma$ concentrations that activate GIRK and regulate other targets of $G\beta\gamma$. The higher rate of $G\beta\gamma$ release is attributable to a faster G protein coupled receptor – G protein trimer association rate in M2R compared to $\beta2AR$. Thus, a rate difference in a single kinetic step accounts for specificity.

DOI: https://doi.org/10.7554/eLife.42908.001

## Introduction

Heart rate is tightly regulated by the combined effects of the sympathetic and parasympathetic branches of the autonomic nervous system. These two branches control heart rate by stimulating different G protein-coupled receptors (GPCRs), which in turn activate ion channels that modify the electrical properties of cardiac pacemaker cells (*DiFrancesco, 1993*). Sympathetic stimulation accelerates heart rate through activation of beta-adrenergic receptors ($\beta ARs$) and the stimulatory G protein ($G\alpha_s$) pathway, while parasympathetic stimulation slows heart rate through activation of the muscarinic acetylcholine receptor $M_2$ (M2Rs) of the inhibitory G protein ($G\alpha_i$) pathway (*Brodde and Michel, 1999*; *Gordan et al., 2015*).

G protein-activated inward rectifier $K^+$ (GIRK) channels are targeted by the parasympathetic nervous system (*Loewi, 1921*; *Irisawa et al., 1993*; *Schmitt et al., 2014*). Upon stimulation, acetylcholine (ACh) released from the vagus nerve binds to and activates M2Rs in sinoatrial node (SAN) pacemaker cells, promoting the engagement of the GDP-bound G protein trimer ($G\alpha_i(GDP)\beta\gamma$). The activated receptor catalyzes removal of GDP from the G protein alpha subunit ($G\alpha_i$), which allows intracellular GTP to bind. The GTP-bound $G\alpha$ ($G\alpha_i(GTP)$) and the G protein beta-gamma subunit ($G\beta\gamma$) then dissociate from the receptor and from each other (*Figure 1A*) (*Hilger et al., 2018*). The $G\beta\gamma$ subunit, now free to diffuse on the intracellular membrane surface (attached by a lipid anchor), binds to GIRK and causes it to open (*Sakmann et al., 1983*; *Soejima and Noma, 1984*; *Logothetis et al., 1987*; *Wickman et al., 1994*; *Krapivinsky et al., 1995*). Open GIRK channels hyperpolarize the cell membrane and thus lengthen the interval between cardiac action potentials (i.e. slow the heart rate) (*DiFrancesco, 1993*; *Schmitt et al., 2014*). The process is reversed by the alpha subunit, which hydrolyses GTP to GDP followed by reformation of the $G\alpha_i(GDP)\beta\gamma$ complex.

Sympathetic stimulation of $\beta AR$ speeds heart rate by opening excitatory ion channels through the $G\alpha_s$ pathway (*DiFrancesco and Tortora, 1991*; *Simonds, 1999*). Important to balanced opposing effects of sympathetic and parasympathetic input, $\beta AR$ stimulation does not open GIRK channels even though $G\beta\gamma$ subunits are released by this receptor (*Hein et al., 2006*; *Digby et al., 2008*). The reason why GIRK channel opening is specific to $G\alpha_i$-coupled GPCR stimulation and not to $G\alpha_s$-coupled GPCR stimulation has remained a long-standing unsolved puzzle, which we refer to as the $G\beta\gamma$

*For correspondence:
mackinn@mail.rockefeller.edu

Competing interests: The authors declare that no competing interests exist.

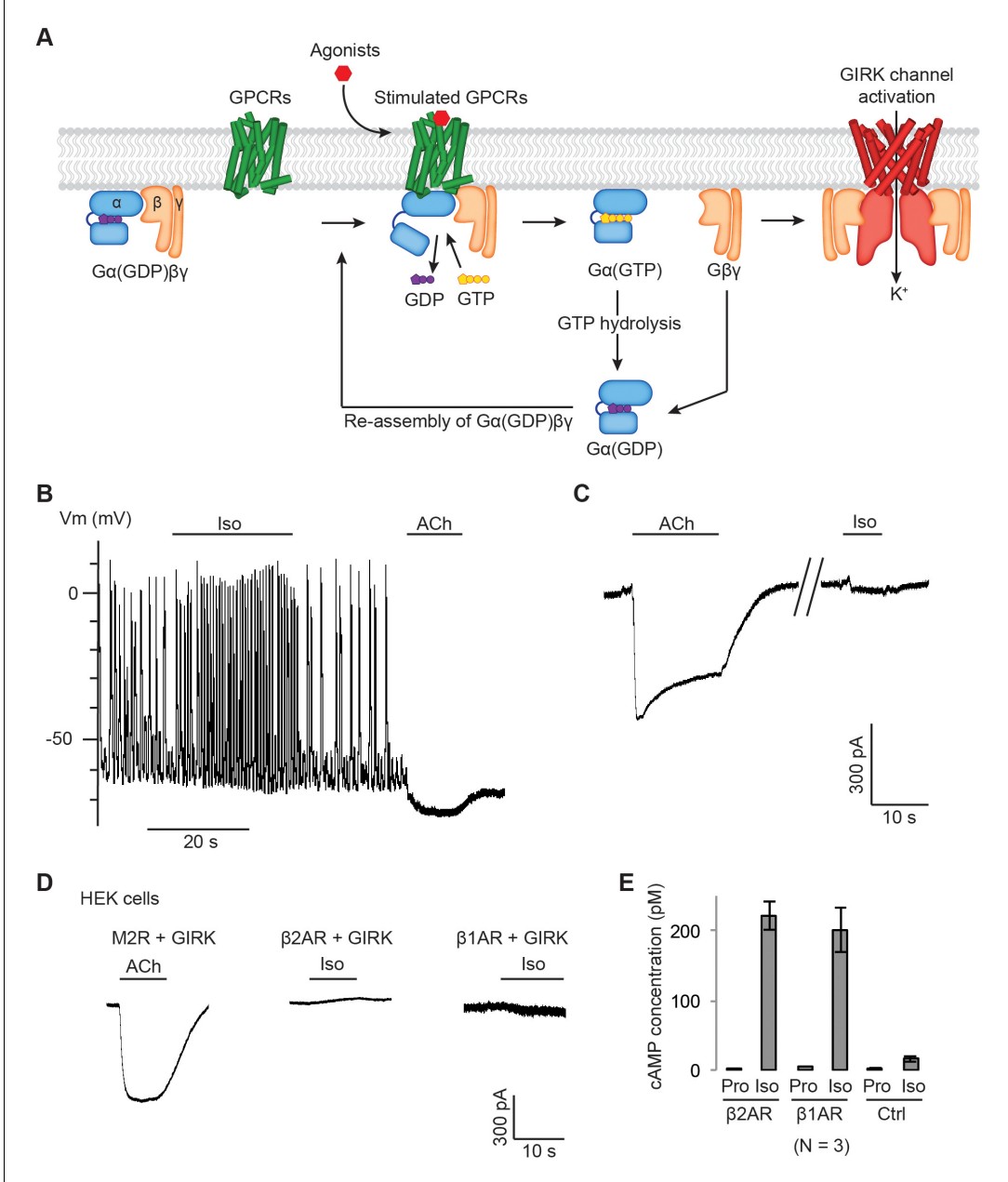

**Figure 1.** Gβγ specificity between GPCRs and GIRK channels. (**A**) A schematic representation of GPCR signal transduction and GIRK channel activation. Agonist binding promotes the formation of a GPCR-Gα(GDP)βγ complex. The activated GPCR then triggers the exchange of GDP to GTP on the Gα subunit. Gα(GTP) and Gβγ subunits subsequently dissociate from the GPCR. Dissociated Gβγ directly binds to and activates GIRK channels. Dissociated Gα(GTP) hydrolyzes GTP to GDP, which then reassociates with Gβγ to form Gα(GDP)βγ. (**B**) A representative current-clamp recording of spontaneous action potentials from an acutely isolated murine sinoatrial node (SAN) cell. 1 μM isoproterenol (Iso) or acetylcholine (ACh) was applied as indicated. (**C**) A representative voltage-clamp recording from the same SAN cell in (**B**). The membrane potential was held at −80 mV, and 1 μM Iso or ACh was applied as indicated. (**D**) Representative voltage-clamp recordings of HEK-293T cells transiently co-transfected with GIRK channels, and either M2Rs, β2ARs or β1ARs. The membrane potential was held at −80 mV. 10 μM ACh or Iso was applied as indicated. (**E**) Validation of the function of βARs. HEK-293T cells expressing βARs or untransfected HEK-293T cells (Ctrl) were treated with 10 μM propranolol (Pro) or isoprennaline (Iso), and intracellular cAMP levels were quantified (N = 3, ±SD). See also *Figure 1—figure supplement 1*.

DOI: https://doi.org/10.7554/eLife.42908.002

The following figure supplement is available for figure 1:

**Figure supplement 1.** Gβγ specificity in heterologously expressed GIRK channels.

DOI: https://doi.org/10.7554/eLife.42908.003

specificity puzzle. One theory posited the existence of a macromolecular super-complex consisting of GIRK, G proteins and a Gα$_i$-coupled GPCR, to endow specificity by proximity (*Peleg et al., 2002*; *Ivanina et al., 2004*; *Clancy et al., 2005*; *Riven et al., 2006*). Another theory suggested that stimulated Gα$_s$-coupled receptors might generate insufficient quantities of free Gβγ if Gα$_s$(GTP) binds to Gβγ with higher affinity (*Digby et al., 2008*). None of these studies provided sufficient data to strongly support a solution. Here we present data that support a simple biochemical solution to this puzzle.

## Results

### Gβγ specificity in native and heterologously expressed GIRK channels

*Figure 1B* shows spontaneous action potentials recorded from current-clamped murine SAN cells isolated from adult mice. With application of isoproterenol (Iso), action potential (AP) frequency increased, and with ACh firing altogether ceased. Somewhat surprisingly, we could not find in the literature a demonstration of both autonomic responses in the same cell. Here we observe that both Gα$_i$-associated (*via* ACh to slow heart rate) and Gα$_s$-associated (*via* Iso to speed heart rate) pathways are indeed activated within a single pacemaker cell. *Figure 1C* shows a voltage-clamp experiment performed on the very same cell shown in *Figure 1B*. ACh produces inward K$^+$ current through GIRK channels, which is the origin of action potential cessation in *Figure 1B*. Iso does not activate GIRK even though βAR stimulation is known to generate free Gβγ subunits.

*Figure 1D* shows voltage clamp experiments in human embryonic kidney 293T (HEK-293T) cells in which GIRK channels and GPCRs were heterologously expressed. M2R is a Gα$_i$-coupled GPCR stimulated by ACh and beta 1-adrenergic receptor (β1AR) and beta 2-adrenergic receptor (β2AR) are both Gα$_s$-coupled GPCRs stimulated by Iso. In each experiment, agonist (ACh or Iso) is applied to reveal the level of stimulated K$^+$ current. Only M2R receptor stimulation activates GIRK to a large extent. This expression is not due to endogenous M2Rs in HEK-293T cells, as ACh fails to stimulate GIRK channels unless M2R is expressed (*Figure 1—figure supplement 1A*). A difference in surface expression levels of the GPCRs does not explain this result, as Alexa Fluor 488-labeled M2Rs and β2ARs show similar fluorescence intensity at the plasma membrane (*Figure 1—figure supplement 1B–1C*). To ensure that expressed β1AR and β2AR are indeed functional in the cells and capable of initiating the Gα$_s$ pathway, the cAMP ELIZA assay was used to measure Iso-stimulated increases in cyclic adenosine monophosphate (cAMP) concentration, which is not observed in control cells and is thus dependent on the β1AR and β2AR expression (*Figure 1E*). Similar experiments were carried out in chinese hamster ovary (CHO) cells (also mammal-derived) and Spodoptera frugiperda (Sf9) cells (insect-derived) (*Figure 1—figure supplement 1D–1E*). In each cell line only M2R receptor stimulation activates GIRK channels. These data demonstrate that specificity persists across mammalian and insect cells and is therefore a robust property of these signaling pathways. The results also imply that GIRK activation does not depend on Gβγ subtypes, because different cell lines, particularly Sf9 cells, express subtypes of Gβγ that are distinct from those in mammals (*Leopoldt et al., 1997*).

### Effect of artificially enforced GPCR-GIRK co-localization

To test whether the macromolecular supercomplex hypothesis can account for Gβγ specificity, we artificially enforced proximity by expressing GIRK linked to either M2R or β2AR within a single open reading frame, as shown (*Figure 2A*). When expressed and analyzed using a western blot, the linked GIRK channel and GPCR run on SDS-PAGE gels as either full-length GIRK-GPCR units or as dimers, trimers and tetramers of those units (*Figure 2B*). Therefore, when expressed, GIRK and the GPCR remain linked together. Because GIRK channels are tetramers under native conditions, expression of the GIRK-GPCR unit causes each channel to be surrounded by four GPCRs. Voltage-clamp experiments on HEK-293T cells transiently transfected with the M2R-GIRK construction showed GIRK activation in response to ACh stimulation (*Figure 2C*). Iso stimulation with cells expressing the β2AR-GIRK construction did not activate GIRK (*Figure 2D*), even though the β2AR is functional as evidenced by quantifying levels of stimulated cAMP (*Figure 2E*). These experiments do not support the macromolecular supercomplex hypothesis as an explanation for Gβγ specificity.

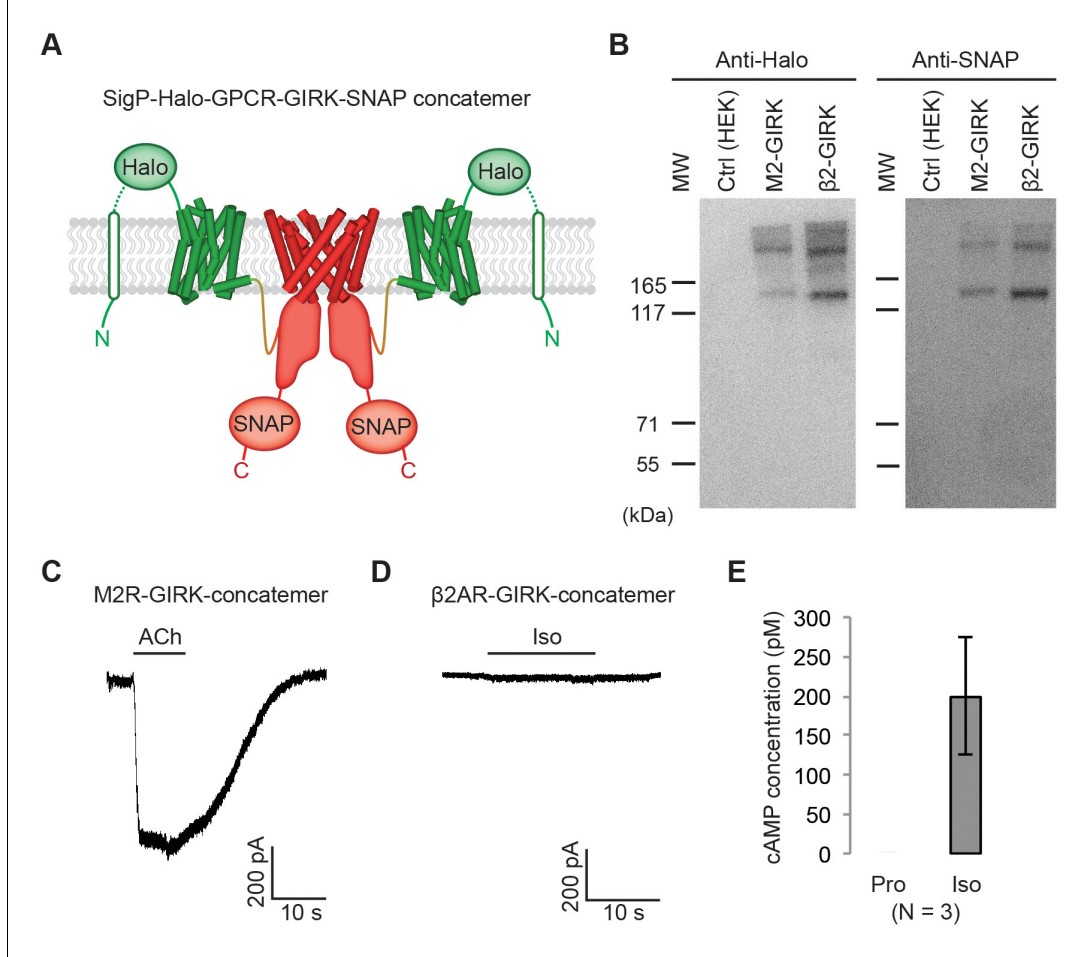

**Figure 2.** Effect of artificially enforced GPCR-GIRK co-localization. (**A**) A schematic representation of GPCR-GIRK concatemer constructs. GIRK was directly fused to the C-terminus of GPCRs. A cleavable signal peptide and a Halo tag were added to the N-terminus of each concatemer. Additionally, a SNAP tag was added to the C-terminus of each concatemer. (**B**) Western-Blot analysis of GPCR-GIRK concatemer constructs. HEK-293T cells were transiently transfected with either M2R-GIRK or β2AR-GIRK concatemers. The expected size of these concatemers is ~150 kDa. (**C**) (**D**) Representative voltage-clamp recordings of HEK-293T cells transiently transfected with M2R-GIRK concatemers or β2AR-GIRK concatemers. Membrane potential was held at −80 mV. 10 μM ACh or Iso was applied as indicated. (**E**) Validation of the function of β2AR-GIRK concatemers. HEK-293T cells expressing β2AR-GIRK concatemers were treated with 10 μM propranolol (Pro) or isoproterenol (Iso), and intracellular cAMP levels were quantified (N = 3, ±SD).
DOI: https://doi.org/10.7554/eLife.42908.004

## Influence of G protein levels on specificity

In the experiments described so far, activation of GIRK channels by GPCR stimulation was facilitated by endogenous levels of G proteins in the cells. We next ask what happens if the levels of G proteins available for mediating activation are altered? Using a cell line in which we established stable expression of GIRK channels and GPCRs, G protein levels were altered using transient transfection. In control experiments endogenous G protein levels support M2R stimulated GIRK channel activation (*Figure 3A*), as was observed in *Figure 1*. Expression of additional $G\alpha_{i1}$ subunits suppressed the level of M2R-stimulated GIRK current, presumably because excess $G\alpha_{i1}$ subunits blunt the normal increase in $G\beta\gamma$ concentration (i.e. $G\alpha_{i1}$ can compete with the channel for available $G\beta\gamma$). Expression of additional $G\alpha_{i1}$ and $G\beta\gamma$ subunits, however, leads to M2R-stimulated GIRK current that exceeds levels mediated by endogenous G proteins alone (*Figure 3A* and *Figure 3—figure supplement 1A*). This latter observation would seem to suggest that increased availability of $G\alpha_i(GDP)\beta\gamma$ substrate (upon which stimulated M2R acts to generate free $G\beta\gamma$) leads to increased $G\beta\gamma$ levels following M2R stimulation. The question then naturally arises, if sufficiently high levels of $G\alpha_s(GDP)\beta\gamma$ substrate are provided, might the β2AR activate GIRK to a detectable extent? The answer is yes.

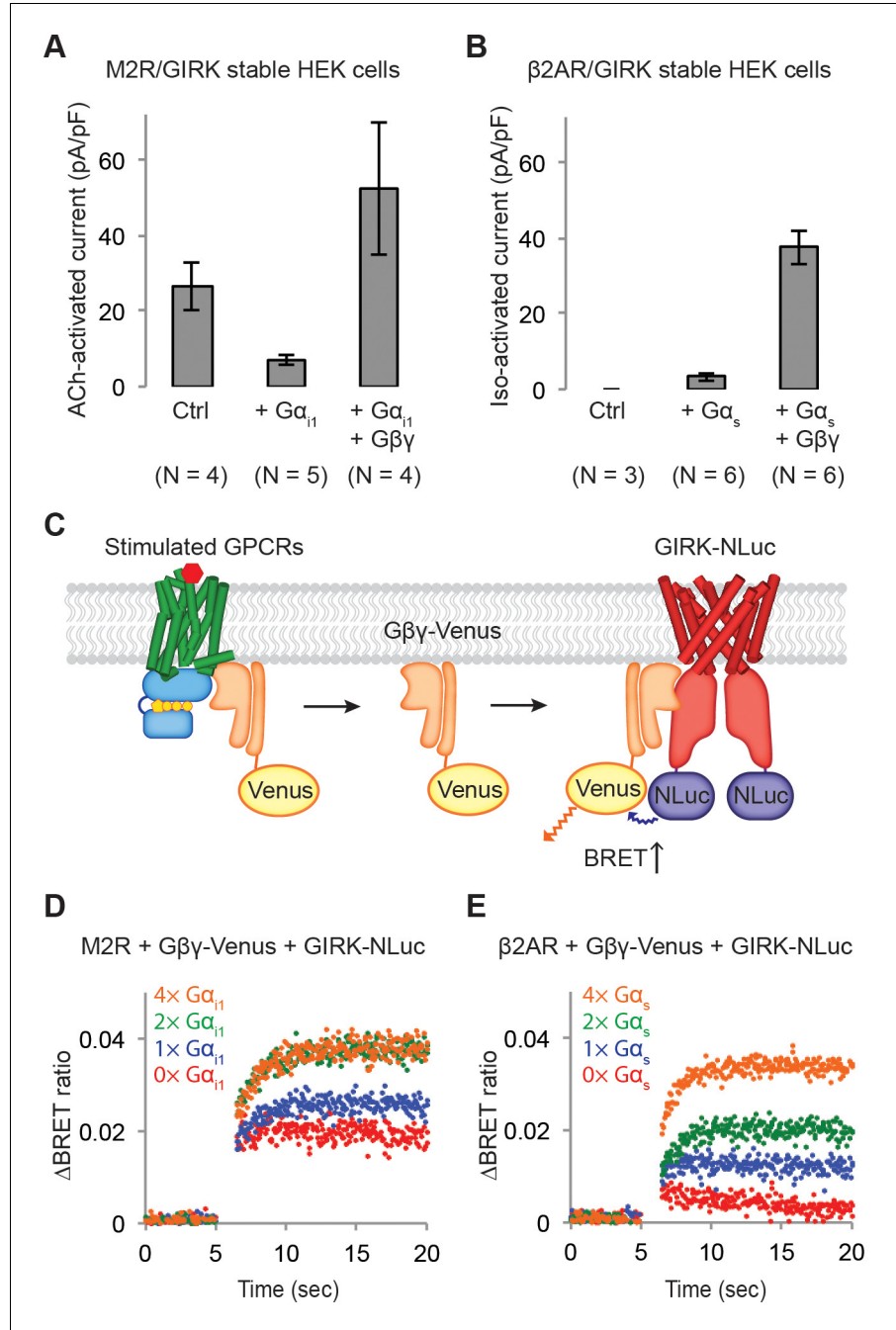

**Figure 3.** Influence of G protein levels on specificity. (**A**) GIRK currents induced by M2R agonist ACh. Cells from a stable HEK-293T cell line expressing M2Rs and GIRK channels were transiently transfected with a vector expressing either GFP (Ctrl), Gα$_{i1}$, or Gα$_{i1}$ and Gβγ. 10 μM ACh was applied, and the evoked inward current was normalized to the capacitance of the cell (±SEM). (**B**) GIRK currents induced by β2AR agonist Iso. Cells from a stable HEK-293T cell line expressing β2ARs and GIRK channels were transiently transfected with a control vector expressing either GFP, Gα$_s$, or Gα$_s$ and Gβγ. 10 μM Iso was applied, and the evoked inward current was normalized to the capacitance of the cell (±SEM). (**C**) A schematic representation of the BRET assay. Upon agonist stimulation of a GPCR, Gβγ-Venus is released. Gβγ-Venus then binds to GIRK-NLuc, which increases the BRET signal. (**D**) (**E**) Representative changes in BRET signal upon stimulation of GPCRs. In (**D**), HEK-293T cells were transfected with M2Rs, Gβγ-Venus, GIRK-NLuc, and increasing amounts of Gα$_{i1}$. In (**E**), HEK-293T cells were transfected with β2ARs, Gβγ-Venus, GIRK-NLuc, and increasing amounts of Gα$_s$. Agonists were applied at t = 5 s. See also *Figure 3—figure supplements 1* and *2*.

DOI: https://doi.org/10.7554/eLife.42908.005

*Figure 3 continued on next page*

*Figure 3 continued*

The following figure supplements are available for figure 3:

**Figure supplement 1.** β2ARs activate GIRK channels in the presence of over-expressed G protein trimers.
DOI: https://doi.org/10.7554/eLife.42908.006

**Figure supplement 2.** Comparison of endogenous Gα levels in HEK-293T cells.
DOI: https://doi.org/10.7554/eLife.42908.007

Experiments using cells expressing GIRK channels and β2ARs show that excess $G\alpha_s$ and $G\beta\gamma$ subunits give rise to β2AR-stimulated GIRK current (*Figure 3B* and *Figure 3—figure supplement 1B*). This finding suggests that the specificity exhibited by $G\alpha_i$-coupled GPCRs versus $G\alpha_s$-coupled GPCRs is somehow related to differences in the levels of $G\beta\gamma$ that they each are able to generate.

## Direct measurement of the Gβγ–GIRK interaction

We explored the influence of G protein levels further using a more direct measurement to estimate the $G\beta\gamma$-GIRK interaction. After fusing the modified yellow fluorescent protein Venus to $G\beta\gamma$ and the bioluminescent protein Nano-Luciferase (NLuc) to GIRK (GIRK-NLuc) we monitored their proximity by measuring the bioluminescent resonance energy transfer (BRET) ratio (*Masuho et al., 2015*). The idea is, following GPCR stimulation $G\beta\gamma$-Venus separates from the GPCR-G protein complex and binds to GIRK, bringing Venus close to NLuc on the channel and thus increasing the BRET ratio (*Figure 3C*).

Two initial controls were carried out. First, we examined the binding of $G\beta\gamma$-Venus to the membrane anchored C-terminal PH domain of GRK3 fused to NLuc (masGRK3ct-NLuc), which is known to bind to $G\beta\gamma$ with ~20 nM affinity (*Pitcher et al., 1992*). This experiment produced a robust increase in the BRET signal following M2R stimulation (*Figure 3—figure supplement 1C*). Second, we examined the binding of $G\beta\gamma$-Venus to Kir2.2 fused to NLuc. Kir2.2 is structurally similar to GIRK but does not bind to $G\beta\gamma$. No change in BRET signal occurred following M2R stimulation (*Figure 3—figure supplement 1D*). These positive and negative controls imply that the BRET assay may be suitable for monitoring a specific interaction between GIRK and $G\beta\gamma$ subunits released following GPCR stimulation.

HEK-293T cells were transiently transfected with M2Rs, $G\beta\gamma$-Venus, GIRK-NLuc, and varying concentrations of $G\alpha_{i1}$. The BRET signal was then monitored over time following ACh stimulation (*Figure 3D* and *Figure 3—figure supplement 1E*). Even in the absence of additional $G\alpha_{i1}$, the BRET signal showed a time-dependent increase, consistent with $G\beta\gamma$-Venus being released from M2Rs and then binding to the GIRK channel. As the amount of $G\alpha_{i1}$ expression was increased the BRET signal increased further, consistent with more $G\beta\gamma$-Venus being generated as a result of greater $G\alpha_{i1}(GDP)\beta\gamma$-Venus substrate availability. Note that this result is not inconsistent with the reduced current generated in *Figure 3A* upon excess $G\alpha_{i1}$ expression because in the BRET experiment (*Figure 3D*) $G\beta\gamma$-Venus is also over-expressed. As the level of $G\alpha_{i1}$ expression is increased a maximum BRET signal is reached, suggesting that an aspect of this signaling pathway other than $G\alpha_{i1}$ availability eventually becomes limiting. When the same experiment was carried out with the β2AR only a very small change in the BRET signal was observed in the absence of $G\alpha_s$ transfection (*Figure 3E* and *Figure 3—figure supplement 1E*), consistent with the failure of β2AR stimulation (in the absence of $G\alpha_s$ transfection) to activate GIRK channels (*Figure 3B*). In accord with the ability of $G\alpha_s$ and $G\beta\gamma$ over-expression to over-ride specificity and permit β2AR-stimulated GIRK current (*Figure 3B*), the BRET ratio increased with increased expression of $G\alpha_s$ (and $G\beta\gamma$-Venus). The electrophysiological and BRET assays are in complete agreement with each other and suggest that specificity in $G\alpha_i$-coupled GPCR signaling results from higher $G\beta\gamma$ concentrations achieved when $G\alpha_i$-coupled receptors are stimulated compared to $G\alpha_s$-coupled receptors.

## Generalization of Gαi-coupled GPCR target specificity

If specificity results from higher levels of $G\beta\gamma$ generated when $G\alpha_i$-coupled receptors are stimulated rather than from a specific protein-protein interaction and localization of the receptor with GIRK, then other targets upon which $G\beta\gamma$ acts might also exhibit similar specificity. To test this idea, we carried out experiments using the transient receptor potential melastatin 3 (TRPM3) channel, which

is inhibited by Gβγ (*Figure 4A*) (*Badheka et al., 2017*; *Quallo et al., 2017*; *Dembla et al., 2017*). TRPM3 channels and M2Rs were transiently transfected into HEK-293T cells and whole-cell voltage-clamp recordings were performed. TRPM3 channels were first activated by a chemical ligand, pregnenolone sulphate (PS), and then inhibited (85 ± 10%) by stimulating M2R with Ach (*Figure 4B and C*). Similar experiments with Iso-stimulated β2ARs showed only modest inhibition (17 ± 10%), consistent with some degree of specificity as a result of there being insufficient concentrations of Gβγ generated by the Gα$_s$-coupled pathway (*Figure 4B and D*). As in the GIRK experiments, specificity is lost when Gα$_s$ and Gβγ are over-expressed (inhibition 73 ± 14%) (*Figure 4B–4F*). These observations further strengthen the idea that Gα$_i$-coupled receptors generate higher concentrations of Gβγ in the setting of endogenous G protein concentrations and that these higher Gβγ levels account for Gβγ specificity. These observations also further reject the macromolecular supercomplex hypothesis as a tenable explanation, because similar Gβγ specificity is observed with a completely different protein target of the Gβγ pathway.

## Relative rates of Gβγ release by Gα$_i$ versus Gα$_s$-coupled receptors

By what mechanisms do M2Rs generate higher Gβγ concentrations than β2ARs? If Gα$_i$ subunits were more abundant in cells than Gα$_s$ subunits then higher rates of Gβγ generation would be expected. This explanation seems unlikely though, because the endogenous levels of Gα$_s$ in HEK-293T cells are actually higher than Gα$_i$ when we measure levels directly using a Western blot assay in the same cells

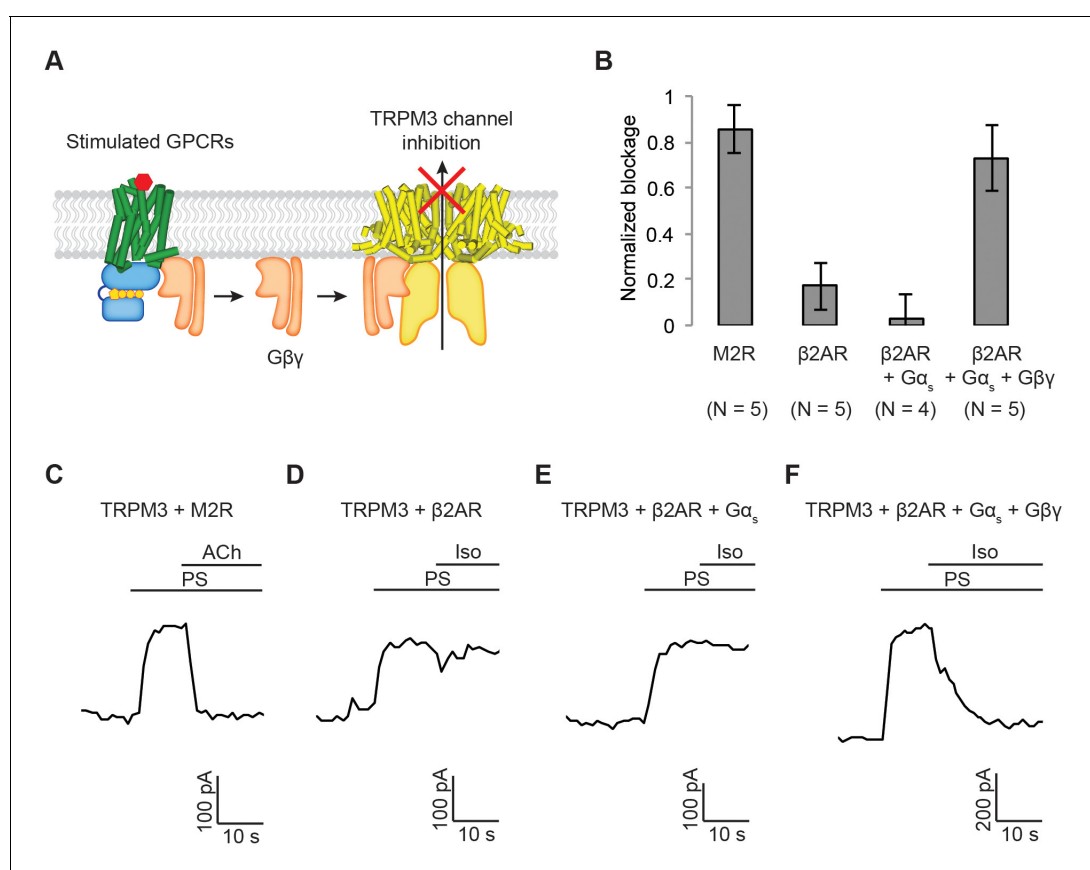

**Figure 4.** Generalization of Gα$_i$-coupled GPCR target specificity. (**A**) A schematic representation of TRPM3 channel inhibition by Gβγ. Upon agonist stimulation, released Gβγ directly binds to and inhibits TRPM3 channels. (**B**) The amount of current blocked upon GPCR stimulation was normalized to the first peak current (±SEM). (**C**)-(**F**) Representative voltage-clamp recordings of HEK-293T cells transiently transfected with (**C**) TRPM3 and M2Rs (**D**) TRPM3 and β2ARs (**E**) TRPM3, β2ARs, and Gα$_s$, or (**F**) TRPM3, β2ARs, Gα$_s$, and Gβγ. A ramp protocol from −100 mV to +100 mV was applied to the cells every second. The currents at +100 mV were plotted. TRPM3 currents were evoked by 10 µM pregnenolone sulfate (PS). M2Rs and β2ARs were stimulated by 10 µM ACh and Iso, respectively.
DOI: https://doi.org/10.7554/eLife.42908.008

(*Figure 3—figure supplement 2*). Higher levels of Gα$_s$ in HEK-293T cells were also reported previously on the basis of RNA levels (*Atwood et al., 2011*).

Alternatively, differences in the affinity of Gβγ for Gα$_s$-GTP versus Gα$_i$-GTP could potentially account for differences in the levels of free Gβγ generated during β2AR versus M2R stimulation. To test this possibility, we assessed the relative ability of Gα$_s$-GTP versus Gα$_{i1}$-GTP to bind to Gβγ. Because the affinity of GTP-bound forms of Gα for Gβγ are so low we contrived the experiment shown in *Figure 5A*. GIRK channels and Gβγ were reconstituted into planar lipid bilayers at a mass ratio of ~1:0.1. In the presence of 8 mM Na$^+$ and 32 μM C8-PIP$_2$ a fraction of GIRK channels are activated in the context of limiting Gβγ concentration (*Figure 5—figure supplement 1A*). Under this condition, sufficiently high concentrations of Gα(GTP-γS) can inhibit GIRK activation through competition by binding to Gβγ. Thus, known amounts of Gα$_{i1}$(GTP-γS) or Gα$_s$(GTP-γS) were added by replacing the lipid tail with a His$_{10}$ tag and including in the bilayer 3% Ni-NTA lipids. After saturation of Ni-NTA lipids these conditions should yield a Gα(GTP-γS) concentration adjacent to the membrane ~3 mM (*Wang et al., 2016*; *Touhara et al., 2016*). Inhibition of GIRK current was observed, but with no significant difference between Gα$_{i1}$(GTP-γS) and Gα$_s$(GTP-γS), suggesting that their affinities for Gβγ are similar (*Figure 5B–5D*). Thus, lower Gβγ concentrations following Gα$_s$-coupled receptor stimulation cannot be attributed to sequestration by Gα$_s$(GTP).

Next, we tested the possibility that Gα$_i$-coupled receptors catalyze intrinsically faster Gβγ release. We developed an assay by attaching Venus to Gα, NLuc to Gβγ, and measured the BRET ratio

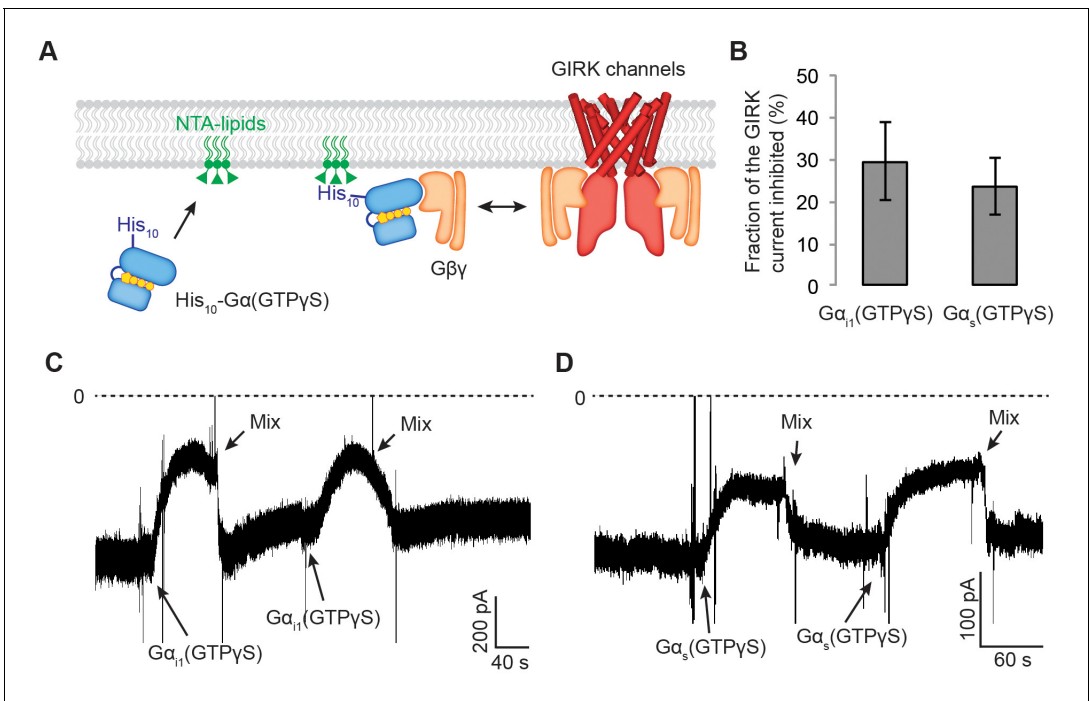

**Figure 5.** Gα$_s$(GTP-γS) and Gα$_{i1}$(GTP-γS) do not differentially compete with GIRK channels for Gβγ. (**A**) A schematic representation of the competition assay between His$_{10}$-Gα(GTP-γS) and GIRK for Gβγ in a reconstituted planar lipid bilayer system. In these experiments, we controlled the amount of lipid-associated Gα(GTP-γS) to evaluate the competition quantitatively. We first incorporated a fixed amount of Ni-NTA-lipids into the lipid bilayer and applied enough His$_{10}$-Gα(GTP-γS) to saturate all the available Ni-NTA binding sites (*Figure 5—figure supplement 1B–1D*). Tethered His$_{10}$-Gα(GTP-γS) competes with GIRK for Gβγ and therefore inhibits GIRK. (**B**) Current inhibition by His$_{10}$-Gα(GTP-γS) was normalized to the initial current levels (N = 3, ±SD). (**C**) (**D**) Representative inward GIRK currents from lipid bilayers. GIRK was partially activated by PIP$_2$, Na$^+$, and a low concentration of Gβγ. Dashed lines represent the baseline current (0 pA). (**C**) His$_{10}$-Gα$_{i1}$(GTP-γS) or (**D**) His$_{10}$-Gα$_s$(GTP-γS) was directly perfused to the bilayer membrane several times followed by mixing the solutions in the bilayer chamber. The transient decrease in the current upon Gα(GTP-γS) application is an artifact due to the absence of Na$^+$ in His$_{10}$-Gα(GTP-γS) solution. See also *Figure 5—figure supplement 1*.

DOI: https://doi.org/10.7554/eLife.42908.009

The following figure supplement is available for figure 5:

**Figure supplement 1.** Purified His$_{10}$-Gα(GTP-γS) binds to the GUV membrane containing Ni-NTA lipids.
DOI: https://doi.org/10.7554/eLife.42908.010

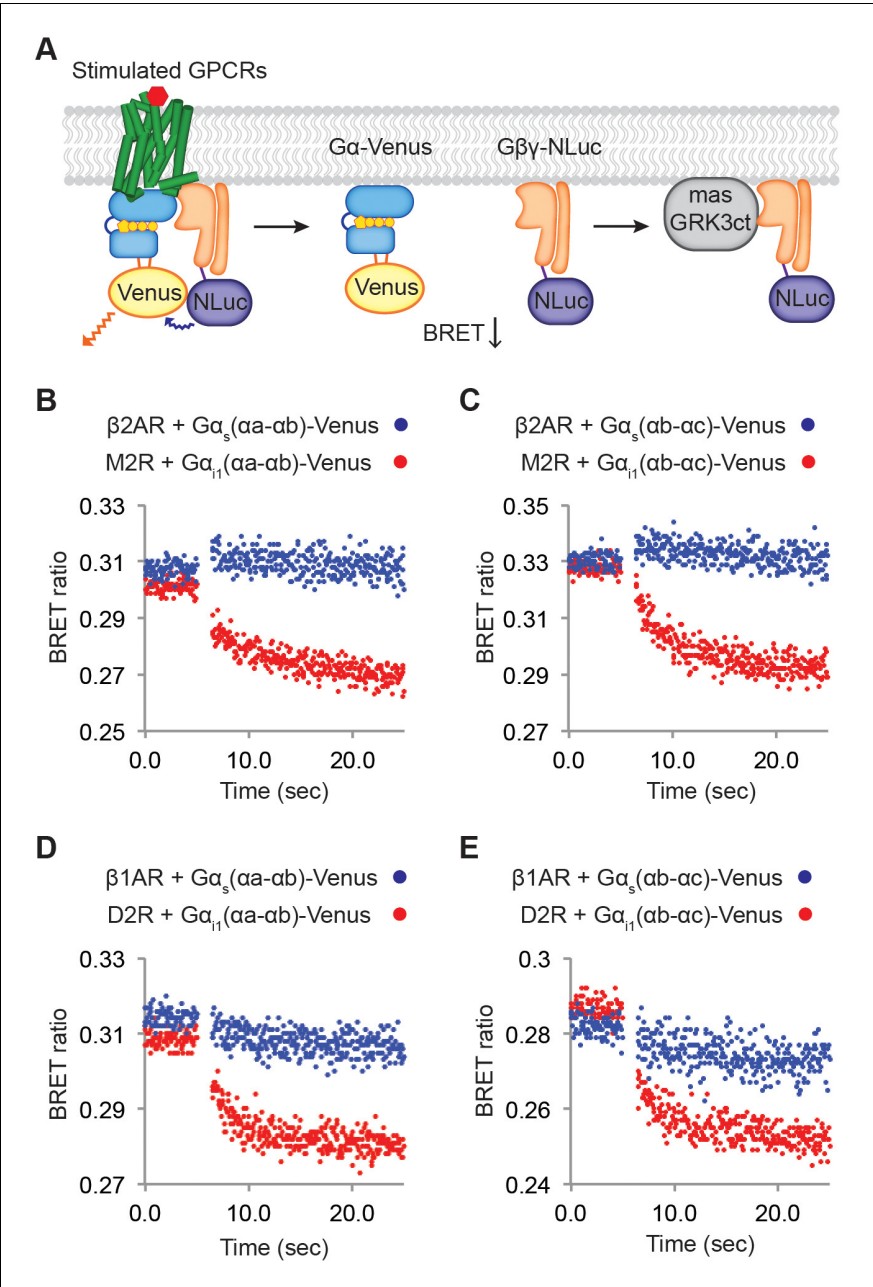

**Figure 6.** Relative rates of Gβγ release by Gα$_i$ versus Gα$_s$-coupled receptors. (**A**) A schematic representation of the experiment that monitors Gβγ release by BRET. Upon agonist stimulation, GPCRs release Gα and Gβγ. The dissociation of Gα-Venus and Gβγ-NLuc results in a decrease of the BRET signal. Released Gβγ-NLuc was chelated by masGRK3ct, a fusion of the C-terminal PH domain of GRK3 and a myristic acid attachment peptide. (**B**)-(**E**) Representative time-resolved BRET ratio curves. In (**B**) and (**C**), M2Rs released more Gβγ than β2ARs did within the same time period, independent of which Gα-Venus construct was used. In (**D**) and (**E**), D2Rs released more Gβγ than β1ARs did within the same time period, independent of which Gα-Venus construct was used. See also *Figure 6—figure supplements 1* and *2*, and *Table 1*.

DOI: https://doi.org/10.7554/eLife.42908.011

The following figure supplements are available for figure 6:

**Figure supplement 1.** M2Rs catalyze release of Gβγ at higher rate compared to β2ARs.

DOI: https://doi.org/10.7554/eLife.42908.012

**Figure supplement 2.** D2Rs catalyze release of Gβγ at higher rate compared to β1ARs.

DOI: https://doi.org/10.7554/eLife.42908.013

change to monitor GPCR-mediated dissociation of Gβγ-NLuc from Gα-Venus (*Figure 6A*). We also expressed masGRK3ct in the same cells to sequester Gβγ-NLuc once it is released, thus reducing the extent to which Gβγ-NLuc will rebind to Gα-Venus. Two different Gα-Venus insertion constructs were made – into the αa-αb loop or into the αb-αc loop of Gα – to ensure that the observed behavior does not depend on the site of insertion (*Figure 6—figure supplement 1A*). Prior to GPCR stimulation, N-Luc intensity and BRET ratio were nearly constant and approximately similar in magnitude in all experiments (*Figure 6B–6C*, *Figure 6—figure supplement 1*, and *Table 1*). Following GPCR stimulation the BRET ratio change was small for the β2AR and comparatively large for M2R. Similar experiments were also carried out with the Gα$_i$-coupled dopamine receptor (D2R), which activates GIRK (*Figure 6—figure supplement 2A*), and the Gα$_s$-coupled β1AR, which does not. Again, we observe that Gβγ-dissociation from Gα is much greater for the Gα$_i$-coupled receptor (*Figure 6D–6E*, *Figure 6—figure supplement 2C–2D*, and *Table 1*). The small signal associated with βAR stimulation is not due to malfunctioning of the Venus-inserted Gα$_s$ constructs because Gβγ-NLuc dissociation from Gα$_s$-Venus is observable in controls in which G proteins were over-expressed to higher levels (i.e. when BRET ratio prior to βAR stimulation was higher) (*Figure 6—figure supplement 2B*). We conclude from these experiments that the Gα$_i$-coupled receptors M2R and D2R generate more rapid Gβγ release than the Gα$_s$-coupled receptors β1AR and β2AR due to a higher intrinsic turnover rate.

## Kinetic model of Gβγ specificity

We developed a kinetic model for GIRK activation to test whether we could replicate Gβγ-specificity on the basis of differences in Gα$_i$ versus Gα$_s$-coupled receptor turnover rates. The model consists of a G protein turnover reaction cycle and a GIRK-Gβγ binding reaction that leads to channel activation (*Figure 7A*). Numerous studies have provided estimates for rate constants in the reaction cycle (*Table 2*) (*Breitwieser and Szabo, 1988*; *Sarvazyan et al., 1998*; *Sungkaworn et al., 2017*), and the GIRK-Gβγ binding reaction has been studied in detail, providing estimates for $k_{56}$ and $k_{65}$ as well as a cooperativity factor μ (*Shea et al., 1997*; *Wang et al., 2016*; *Touhara et al., 2016*).

The G protein reaction cycle models the conversion of Gα(GDP)βγ (the G protein trimer) into Gα (GTP) and Gβγ in two kinetic transitions. The first transition ($k_{12}$) describes the formation of a productive complex between the G protein trimer and an active (ligand-bound) GPCR (R*). The second ($k_{23}$) combines multiple reactions, including GDP/GTP exchange and Gα(GTP) and βγ dissociation. In our experiments, the observation that G protein over-expression increases levels of stimulated Gβγ in cells (*Figure 3* and *Figure 4B*) implies that the $k_{12}$ transition is to some extent rate-limiting under physiological G protein conditions. A single molecule study of the α2 adrenergic receptor (α2AR; a Gα$_i$-coupled GPCR) also concluded that complex formation between activated receptor and G protein trimer (i.e. the $k_{12}$ transition) was rate-limiting (*Sungkaworn et al., 2017*). Furthermore, the same study found that $k_{12}$ for the β2AR was ten times smaller than for the α2AR.

GPCR density over the entire membrane of atrial cardiac myocytes and in CHO cells is approximately 5 μm$^{-2}$ (*Nenasheva et al., 2013*). However, G protein signaling occurs within 'hotspots' that

**Table 1.** Quantitative-BRET measurements of Gβγ release from different Gα constructs.
Averaged Nano-Luc intensity, basal BRET ratio, and ΔBRET ratio were summarized (N = 3–4, ±SD).

|  | NLuc intensity | Basal BRET ratio | ΔBRET ratio |
|---|---|---|---|
| M2R-Gα$_{i1}$(αa-αb) | (13.85 ± 0.36) x10⁵ | 0.301 ± 0.003 | 0.031 ± 0.002 |
| β2AR- Gα$_s$(αa-αb) | (11.91 ± 2.80) x10⁵ | 0.310 ± 0.007 | 0.002 ± 0.006 |
| M2R-Gα$_{i1}$(αb-αc) | (8.08 ± 1.77) x10⁵ | 0.310 ± 0.014 | 0.029 ± 0.006 |
| β2AR- Gα$_s$(αb-αc) | (8.64 ± 2.62) x10⁵ | 0.311 ± 0.014 | 0.005 ± 0.005 |
| D2R-Gα$_{i1}$(αa-αb) | (23.29 ± 1.46) x10⁵ | 0.312 ± 0.005 | 0.028 ± 0.006 |
| β1AR- Gα$_s$(αa-αb) | (24.51 ± 1.09) x10⁵ | 0.315 ± 0.007 | 0.007 ± 0.003 |
| D2R-Gα$_{i1}$(αb-αc) | (5.58 ± 4.31) x10⁵ | 0.287 ± 0.001 | 0.032 ± 0.004 |
| β1AR- Gα$_s$(αb-αc) | (7.77 ± 0.34) x10⁵ | 0.279 ± 0.004 | 0.008 ± 0.003 |

DOI: https://doi.org/10.7554/eLife.42908.014

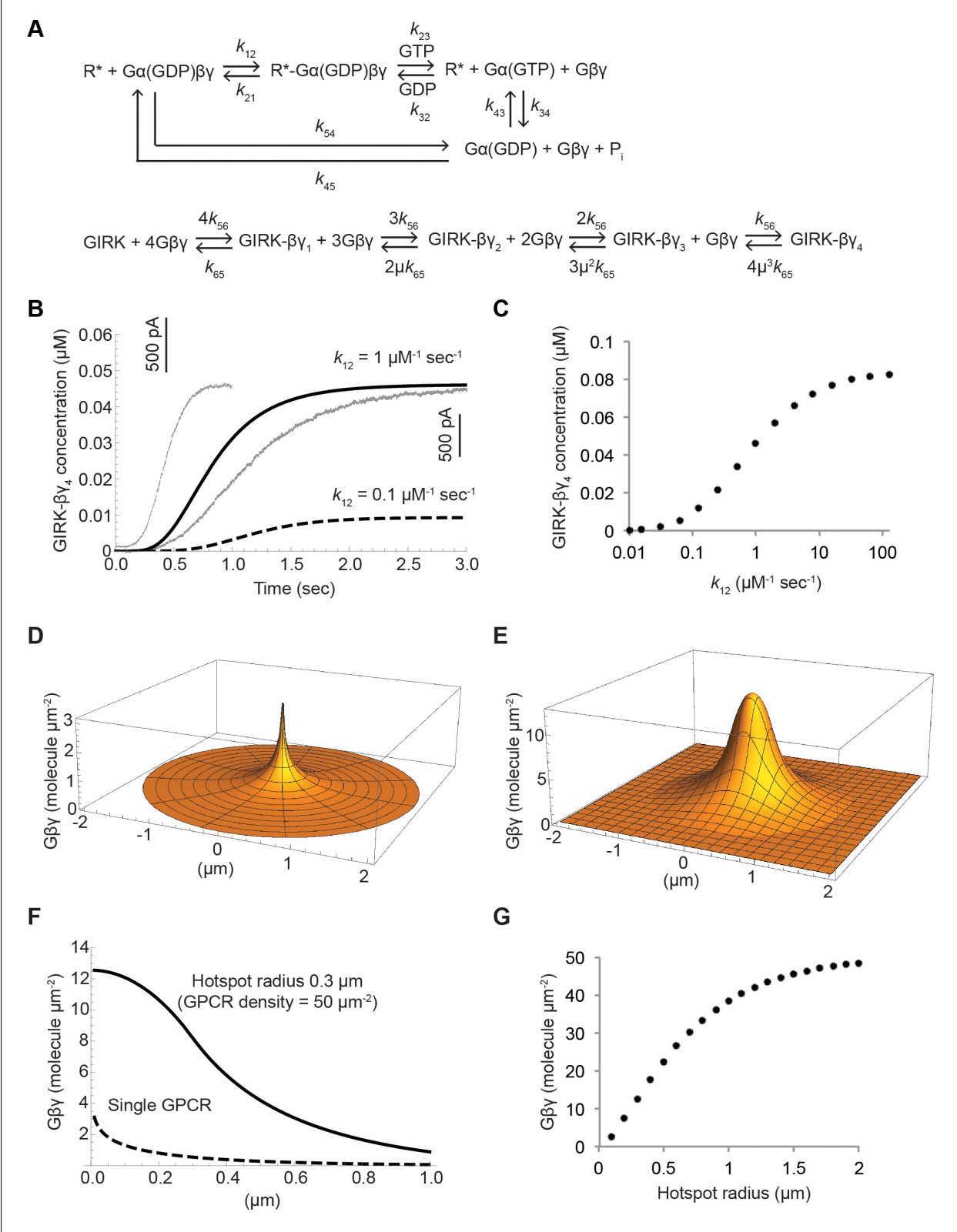

**Figure 7.** Kinetic model of Gβγ specificity. (**A**) Reaction scheme used to model GPCR activation of GIRK. $k_{xy}$ are the rate constants of the reactions between two G protein states. Rate, equilibrium and cooperativity constants are summarized in *Table 2*. (**B**) ACh-stimulated GIRK currents from two different SAN cells are shown in grey. Calculated GIRK-βγ$_4$ concentration as a function of time for two different $k_{12}$ magnitudes are shown in black solid and dashed curves. (**C**) Calculated steady state GIRK-βγ$_4$ concentration as a function of $k_{12}$ magnitude. (**D**) Steady state two-dimensional Gβγ

*Figure 7 continued on next page*

*Figure 7 continued*

concentration profile (molecules $\mu m^{-2}$; one molecule $\mu m^{-2}$ = 0.2 $\mu M$ in a layer 80 Å thick below the membrane surface) surrounding a single GPCR generating 1 Gβγ $sec^{-1}$ with mean Gβγ lifetime 1 s and diffusion coefficient 0.2 $\mu m^2$ $sec^{-1}$. (E) Steady state two-dimensional concentration profile of Gβγ in and surrounding a hotspot of radius 0.3 $\mu m$ with a density of 50 GPCR $\mu m^{-2}$. Gβγ lifetime and diffusion coefficient are the same as in (D). (F) Two dimensional cross sections of concentration profiles in (D) and (E). (G) Steady state Gβγ concentration at the center of hotspot as a function of hotspot radius. See also *Figure 7—figure supplement 1*, and *Table 2*.

DOI: https://doi.org/10.7554/eLife.42908.015

The following figure supplement is available for figure 7:

**Figure supplement 1.** Simulation of GPCR-activation of GIRK.

DOI: https://doi.org/10.7554/eLife.42908.016

we estimate to cover about 10% of the membrane surface (*Sungkaworn et al., 2017*). Thus, we assume the receptor density to be 50 $\mu m^{-2}$ within a hotspot and assume an initial Gα(GDP)βγ density of 100 $\mu m^{-2}$. When the reaction is switched on (i.e. ligand stimulation) at t = 0 by changing $k_{12}$ from 0 to 0.2 $\mu m^2$ molecule$^{-1}$ sec$^{-1}$ (*Sungkaworn et al., 2017*), Gβγ concentration increases (along with time-dependent concentration changes of other components) and GIRK channels activate to a steady state value within a few seconds following a time course similar to M2R stimulated GIRK currents in SAN cells (*Figure 7B* and *Figure 7—figure supplement 1A*). We note that time courses vary from cell to cell, but that the modeled time course falls within the experimental range.

To model the β2AR receptor we reduced $k_{12}$ ten times, consistent with Sungkaworn et al, leaving all other quantities the same. Lower concentrations of Gβγ are predicted and along with significantly less GIRK activation (*Figure 7B*). *Figure 7C* displays in greater detail calculated GIRK-(Gβγ)$_4$ concentration (i.e. channel activation) as a function of $k_{12}$ magnitude. A steep dependence occurs right around the experimentally determined value for the Gα$_i$-coupled receptor turnover rate constant (*Sungkaworn et al., 2017*). Thus, the model predicts that higher rates of G protein turnover catalyzed by Gα$_i$-coupled compared to Gα$_s$-coupled GPCRs can account for Gβγ specificity.

Partial agonists by definition activate GPCRs with reduced efficacy compared to full agonists (*McKinney et al., 1991*). The effects of two partial agonists, oxotremorine and pilocarpine, on M2R activation of GIRK are shown (*Figure 7—figure supplement 1B–1C*). A study recently concluded that for the β2AR, the distinction between partial and full agonist action lies in the magnitude of $k_{12}$, its value being smaller for partial agonists (*Gregorio et al., 2017*). We think this conclusion likely applies to M2R as well, based on the following observations. When the partial agonists oxotremorine (Oxo) and pilocarpine (Pilo) are used to stimulate M2R, reduced GIRK currents are associated with reduced BRET signals for Gβγ-Venus binding to GIRK-NLuc (blue symbols in *Figure 7—figure*

**Table 2.** Parameters used for simulation of GPCR-activation of GIRK.

$k_{12}$: The rate of formation of the productive GPCR-G protein complex (*Sungkaworn et al., 2017*). $k_{21}$: The rate of dissociation of the productive GPCR-G protein complex (*Sungkaworn et al., 2017*). $k_{23}$: The rate of nucleotide exchange and subsequent dissociation of GPCRs, Gα(GTP), and Gβγ (*Sungkaworn et al., 2017*). $k_{32}$: The rate of the reverse reaction of nucleotide exchange and dissociation of GPCRs and G proteins. $k_{34}$: The rate of GTP hydrolysis, based on *Breitwieser and Szabo, 1988*. $k_{43}$: The rate of the reverse reaction of GTP hydrolysis. $k_{45}$: The on-rate between Gα(GDP) and Gβγ, adapted from *Sarvazyan et al., 1998*. $k_{54}$: The off-rate between Gα(GDP) and Gβγ, calculated based on $k_{45}$ and $K_d$ = 3 nM (*Sarvazyan et al., 1998*). $k_{56}$: The on-rate between the GIRK and Gβγ is diffusion limited (*Wang et al., 2016*). $k_{65}$: The off-rate between the GIRK and Gβγ were calculated based on $k_{56}$ and our previous $K_d$ measurement (*Shea et al., 1997*).

| Reaction | Forward-rate | Backward-rate | Note |
|---|---|---|---|
| R* + Gα(GDP)βγ ⇌ R*-Gα(GDP)βγ | 1 $\mu M^{-1}$ sec$^{-1}$ ($k_{12}$) | 1 sec$^{-1}$ ($k_{21}$) | *Sungkaworn et al., 2017* |
| R*-Gα(GDP)βγ ⇌ R* + Gα(GTP) + Gβγ | 1 sec$^{-1}$ ($k_{23}$) | 0 $M^{-2}$ sec$^{-1}$ ($k_{32}$) | *Sungkaworn et al., 2017* |
| Gα(GTP) ⇌ Gα(GDP) + P$_i$ | 2 sec$^{-1}$ ($k_{34}$) | 0 $M^{-1}$ sec$^{-1}$ ($k_{43}$) | *Breitwieser and Szabo, 1988* |
| Gα(GDP) + Gβγ ⇌ Gα(GDP)βγ | 0.7 × 10$^6$ $M^{-1}$ sec$^{-1}$ ($k_{45}$) | 0.002 sec$^{-1}$ ($k_{54}$) | *Sarvazyan et al., 1998* |
| GIRK-βγ$_{n-1}$ + (5 - n)Gβγ ⇌ GIRK-βγ$_n$ + (4 - n)Gβγ | (5 - n) × 1 × 10$^7$ $M^{-1}$ sec$^{-1}$ ((5 - n) × $k_{56}$) | n × $\mu^{n-1}$ × 600 sec$^{-1}$ (n × $\mu^{n-1}$ × $k_{65}$) | *Shea et al., 1997*; *Wang et al., 2016* |

DOI: https://doi.org/10.7554/eLife.42908.017

*supplement 1D*). Furthermore, when amounts of available $G\alpha_{i1}$ are increased (so that more $G\alpha_{i1}(GDP)\beta\gamma$-Venus can form), the partial agonist Oxo gives rise to a BRET signal as strong as that of ACh (orange symbols in *Figure 7—figure supplement 1D*). A similar effect was also observed with Pilo, although to a lesser extent. These results are explicable on the basis of the G protein trimer-GPCR on-rate determining the efficacy of different agonists. Thus, $k_{12}$ can explain the difference in agonists versus partial agonists as well as the fundamental difference between M2R and βARs with respect to their ability to activate GIRK channels. In the model we present, $k_{12}$, is rate limiting under physiological G protein concentrations, and its magnitude determines differential rates of Gβγ generation.

## Discussion

The essential conclusion of this study is that M2R catalyzes the generation of Gβγ at a higher rate than β2AR, thus achieving higher concentrations of Gβγ to activate GIRK. The concentrations of $G\alpha_s(GTP)$ generated by β2AR are obviously sufficient to stimulate the downstream-amplified $G\alpha_s$ pathway and speed heart rate, but the lower Gβγ levels generated are insufficient to activate GIRK to a great extent. The higher rate of Gβγ generation by M2R likely stems from an intrinsically higher rate of association with G protein trimer. This conclusion is most easily appreciated through careful inspection of *Figure 3D and E* and *Figure 3—figure supplement 1E*, where it is shown that endogenous levels of Gα (in the presence of expressed Gβγ-Venus to detect Gβγ binding to GIRK) permit Gβγ generation by M2R, but not by β2AR. Furthermore, over-expression of Gα and Gβγ increases the rate of Gβγ generation in both cases, but higher levels of Gα expression are needed for the β2AR to reach its maximum rate. Thus, Gβγ specificity is explicable on the basis of a difference in the rate at which M2R and β2AR associate with G protein trimer, M2R being faster.

The forward rate in the first step of the reaction cycle (*Figure 7A*) is $k_{12}$ [G protein trimer] [GPCR]. Therefore, a difference in the rate constant $k_{12}$ or either reactant concentration will change the forward rate. We have four reasons to conclude that the difference lies primarily in a difference in $k_{12}$. First, we have shown in HEK-293T cells that the endogenous concentration of $G\alpha_i$ does not exceed $G\alpha_s$ (*Figure 3—figure supplement 2*) and therefore it is unlikely that $G\alpha_i$ trimer exceeds $G\alpha_s$ trimer. Moreover, we estimate the levels of GPCR density expressed in HEK-293T cells to be similar for M2R and β2AR (*Figure 1—figure supplement 1B and C*). Second, given that endogenous levels of $G\alpha_i$ are not greater than $G\alpha_s$ in HEK-293T cells, the difference in the rate of Gβγ generation (mediated by M2R versus β2AR) as reported by the BRET assay (*Figure 3—figure supplement 1E*) is explicable on the basis of a difference in $k_{12}$. Third, the higher rate of Gβγ dissociation from Gα, mediated by M2R versus β2AR (and also D2R versus β1AR), is most simply explained by a difference in $k_{12}$ (*Figure 6*). Fourth, the recent single molecule study showing an intrinsically larger $k_{12}$ for α2AR (a $G\alpha_i$-coupled GPCR) compared to β2AR is completely consistent with the three reasons enumerated above (*Sungkaworn et al., 2017*). Therefore, we conclude that the basis of specificity we are describing here is explained by a difference in $k_{12}$. It is possible that differences in $G\alpha_i$ versus $G\alpha_s$ concentrations in certain cell types could further contribute to specificity. However, a difference in $k_{12}$ alone can explain specificity.

When G protein trimer associates with a GPCR, both Gα and receptor undergo a series of conformational changes (*Rasmussen et al., 2011*). A chimera Gα subunit containing mostly $G\alpha_{i1}$ amino acids and only 13 C-terminal $G\alpha_s$ amino acids – that engage the receptor – is known to permit βAR activation of GIRK (*Leaney et al., 2000*). This observation suggests that the Gα conformational change, which involves the main body of the Gα subunit, might be more important in determining the rate of G protein trimer-GPCR association.

The M2R-GIRK signaling pathway is characterized by four key features: M2R density over the entire cell membrane is relatively low (~5 $\mu m^{-2}$), M2R turnover rate (i.e. Gβγ generation rate) is slow (maximum rate $k_{23}$ ~1 $sec^{-1}$), Gβγ lifetime is short (~1 s), and the affinity of Gβγ for GIRK is not very high (1.9 mM for the first Gβγ and a cooperativity factor of 0.3 for each successive Gβγ). These features have important consequences. The expected steady state concentration profile of Gβγ surrounding an isolated M2R catalyzing even at its maximum rate ($k_{23}$) shows that Gβγ never reaches sufficient levels to activate GIRK (*Figure 7D and F*). This is because as Gβγ is generated it both diffuses away (diffusion coefficient, D ~ 0.2 $\mu m^2 sec^{-1}$) and is re-sequestered by Gα(GDP) in approximately one second (k ~ 1 $sec^{-1}$), causing the Gβγ concentration to decay over a characteristic

distance $(D/k)^{1/2}$. This circumstance explains why β2AR does not activate GIRK even when it is tethered to the channel (*Figure 2*). And it implies that the macromolecular super-complex hypothesis can not work very well to activate GIRK or to explain Gβγ specificity. At a density of 5 μm$^{-2}$, M2Rs are too far apart from each other to build up the Gβγ concentration. But at a density of 50 M2Rs μm$^{-2}$, sufficiently high Gβγ concentrations can be reached: the expected concentration profile surrounding a disk-shaped 'hotspot' of radius 0.3 μm is shown (*Figure 7E and F*). In the middle of the hotspot, which contains about 14 M2Rs, Gβγ concentration reaches 12.5 μm$^{-2}$ (2.5 μM in a layer 80 Å thick beneath the membrane), which is enough to activate GIRK channels that happen to be located within the disk. At a fixed GPCR density (50 μm$^{-2}$), the steady state Gβγ concentration depends on the size of the disk (*Figure 7G*). It is notable that the predicted disk size – several hundred nm to 1 μm – required to achieve sufficiently high concentrations of Gβγ to activate GIRK matches well with G protein signaling hotspots observed in cells (*Sungkaworn et al., 2017*).

In summary, Gβγ specificity is determined by more rapid Gα$_i$(GDP)Gβγ association with M2R compared to Gα$_s$(GDP)Gβγ association with β2AR. A sufficient density of GPCRs is required to achieve GIRK-activating concentrations of Gβγ. This is apparently achieved through the formation of hotspots of higher GPCR and G protein density (*Sungkaworn et al., 2017*). But specificity is explained by the magnitude of a rate constant.

# Materials and methods

## Key resources table

| Reagent type (species) or resource | Designation | Source or reference | Identifiers | Additional information |
|---|---|---|---|---|
| Antibody | Rabbit monoclonal anti Gα$_{i1}$ | Abcam | Cat. #: ab140125 | 1:1000 |
| Antibody | Rabbit monoclonal Anti Gα$_{i2}$ | Abcam | Cat. #: ab157204 | 1:1000 |
| Antibody | Mouse monoclonal anti Gα$_o$ | Santa Cruz Biotechnology | Cat. #: sc-13532 | 1:1000 |
| Antibody | Rabbit polyclonal anti Gα$_{s/olf}$ | Santa Cruz Biotechnology | Cat. #: sc-383 | 1:1000 |
| Antibody | Rabbit polyclonal anti SNAP tag | NEB | Cat. #: P9310S | 1:1000 |
| Antibody | Rabbit polyclonal anti Halo tag | Promega | Cat. #: G9281 | 1:1000 |
| Commercial assay or kit | cAMP ELISA Detection Kit | GenScript | Cat. #: L00460 | |
| Commercial assay or kit | NLuc substrate | Promega | Cat. #: N1110 | 1:50 |
| Cell line (Homo Sapiens) | Flp-In-T-REx-293 | Thermo Fisher | RRID:CVCL_U427 | |
| Cell line (Homo Sapiens) | HEK-293 tsA201 | Sigma | RRID: CVCL_2737 | |
| Cell line (Cricetulus griseus) | Chinese Hamster Ovary Cells | Sigma | RRID: CVCL_0213 | |
| Cell line (Spodoptera frugiperda) | Sf9 cells | Sigma | RRID: CVCL_0549 | |
| Software | Mathematica | Wolfram | SCR_014448 | |

## Experimental model and subject details

### Animals

C57BL/6J (Jackson Labs) male and female adult mice (≥10 weeks old) were used. Animals were kept in cages with a 12:12 hr light/dark cycle and unrestricted access to food and water. All experimental procedures were carried out according to a protocol approved by the Institutional Animal Care and Use Committee (IACUC) of The Rockefeller University (Protocol #16864).

## Cell lines

HEK-293T tsA201 cells were obtained from Sigma and maintained in DMEM (Thermo Fisher) supplemented with 10% Fetal Bovine Serum (FBS, Thermo Fisher) and 1% L-glutamine (Thermo Fisher). Chinese Hamster Ovary cells were obtained from Sigma and maintained in DMEM/F12 (Thermo Fisher) supplemented with 10% FBS and 1% L-glutamine. Sf9 cells were obtained from Sigma and maintained in Grace's Insect medium (Thermo Fisher) supplemented with 10% FBS and Pluronic$^{TM}$ F-68 (Thermo Fisher).

## Methods

### Sinoatrial node (SAN) isolation

Adult mice ($\geq$10 weeks old) were anesthetized with 90–150 mg/kg ketamine and 7.5–16 mg/kg xylazine IP (Sigma-Aldrich). After 5–10 min, when mice stopped responding to tail/toe pinches they were secured in the supine position by gently fixing their forepaws and hindpaws to a pinnable work surface on an animal surgery tray. SAN isolation was performed according to a published procedure (*Sharpe et al., 2016*). A midline skin incision was made from the mid abdomen to the diaphragm with a surgical scissor. The heart was exposed after cutting the diaphragm and holding the sternum with curved serrated forceps. The heart was lifted and dissected out of the thoracic cavity as near as possible to the dorsal thoracic wall. The isolated heart was transferred to a petri dish containing Tyrode's solution (140 mM NaCl, 5.4 mM KCl, 1.2 mM KH$_2$PO$_4$, 1 mM MgCl$_2$, 2 mM CaCl$_2$, 5.5 mM D-glucose, 1 mg/mL BSA, 5 mM HEPES-NaOH [pH 7.4]), and quickly washed several times to remove residual blood. The heart was dissected and the ventricles were removed. The atria were transferred to a silicone dissection dish and pinned through the inferior and superior vena cavae and the right and left atrial appendages. The interatrial septum was exposed by opening the anterior wall. Next, the right atrial appendage was removed and the SAN was isolated by cutting along the cristae terminals.

The isolated SAN was transferred to low-Ca$^{2+}$/Mg$^{2+}$ Tyrode's solution (140 mM NaCl, 5.4 mM KCl, 1.2 mM KH$_2$PO$_4$, 0.5 mM MgCl$_2$, 0.2 mM CaCl$_2$, 5.5 mM D-glucose, 50 mM Taurine, 1 mg/mL BSA, 5 mM HEPES-NaOH [pH 7.4]) and incubated for 5 min at 37°C. Next the SAN was washed with low-Ca$^{2+}$/Mg$^{2+}$ Tyrode's solution twice, transferred to low-Ca$^{2+}$/Mg$^{2+}$ Tyrode's solution with enzymes (0.5 mg/mL Elastase [Worthington], 1.0 mg/mL Type II Collagenase [Worthington], and 0.5 mg/mL Protease xiv [Sigma-Aldrich]), and incubated for 15–20 min at 37°C. Digested tissue was transferred to Kraftbrühe (KB) medium (100 mM K-glutamate, 10 mM K-aspartate, 25 mM KCl, 10 mM KH$_2$PO$_4$, 2 mM MgSO$_4$, 20 mM Taurine, 5 mM Creatine, 0.5 mM EGTA, 20 mM D-glucose, 1 mg/mL BSA, 5 mM HEPES-KOH [pH 7.2]), and gently washed. The tissue was washed two more times with KB medium and cells were dissociated by constant trituration at approximately 0.5–1 Hz for 5–10 min. CaCl$_2$ solution was added stepwise (200 µM, 400 µM, 600 µM, and 1 mM) every 5 min to reach to a final concentration of 1 mM. Subsequently an equal volume of Tyrode's solution was gradually added to the KB solution with dissociated cells. Finally, dissociated cells were centrifuged for 3 min at 150 g, resuspended in Tyrode's solution, and plated onto PDL/Laminin pre-coated glass bottom dishes for ~1 hr prior to electrophysiological recordings.

### Whole-cell voltage clamp recordings on SAN cells

Whole-cell voltage clamp recordings were performed with an Axopatch 200B amplifier in whole-cell mode. The analog current signal was low-pass filtered at 1 kHz (Bessel) and digitized at 20 kHz with a Digidata 1440A digitizer. Digitized data were recorded using the software pClamp. Patch electrodes (resistance 2.0–4.0 MΩ) were pulled on a Sutter P-97 puller (Sutter Instrument Company, Novato, CA) from 1.5 mm outer diameter filamented borosilicate glass. Spontaneous action potential recordings were performed using the amphotericin perforated-patch technique in current-clamp mode without current injection. For voltage-clamp recordings membrane potential was held at −80 mV throughout the experiments and the extracellular solution was exchanged via local perfusion with a 100 µm diameter perfusion pencil positioned adjacent to the cell. The bath solution contained 140 mM NaCl, 5.4 mM KCl, 1 mM CaCl$_2$, 1 mM MgCl$_2$, 10 mM D-glucose, 10 mM HEPES-NaOH (pH 7.4) (~290 mOsm). For the voltage-clamp recordings, the extracellular solution was exchanged to high K$^+$ solution containing 130 mM NaCl, 15.4 mM KCl, 1 mM CaCl$_2$, 1 mM MgCl$_2$, 10 mM D-glucose, 10 mM HEPES-NaOH (pH7.4) (~290 mOsm). The pipette solution contained 9 mM NaCl, 140

mM K-gluconate, 2 mM $MgCl_2$, 1.5 mM EGTA-K, 10 mM HEPES-KOH (pH7.4), 3 mM MgATP, 0.05 mM $Na_2GTP$, 200 µM Amphotericin-B (Sigma-Aldrich) (~310 mOsm).

## Establishment of the stable HEK-293T cell lines

A SNAP tag was fused to the C-terminus of the full-length GIRK4 channel. A serotonin 5-HT signal peptide and a Halo tag were fused to the N-terminus of human full-length M2R or β2AR. Both GIRK4-SNAP and Halo-M2R or Halo-β2AR were cloned into the pcDNA5/FRT/TO vector. An internal ribosome entry site (IRES) sequence was inserted between SNAP-GIRK4 and Halo-GPCR to allow for their simultaneous expression under the same promoter. Stable HEK-293T cell lines were produced using the Flp-In T-REx-293 System according to the manufacturer's protocol (ThermoFisher).

## Whole-cell voltage clamp recordings on HEK-293T and CHO cells expressing GIRK channels

Human M2R, β2AR, D2R, and mouse β1AR were cloned into a pCEH vector for mammalian expression. A serotonin 5-HT cleavable signal peptide and a SNAP tag were inserted into the N-terminus of each receptor (Sero-SNAP-GPCR). The C-terminal GFP-tagged GIRK4 (GIRK4-GFP) was cloned into a pCEH vector. Human G proteins ($G\alpha_{i1}$, $G\alpha_s$, and $G\beta_1$-IRES-$G\gamma_2$) were also cloned into a pCEH vector. Sero-SNAP-GPCR and GIRK4-GFP were transiently transfected to HEK-293T or CHO cells, and cells were incubated at 37°C for 20–24 hr. Stable HEK-293T cell lines expressing Sero-Halo-GPCR and GIRK4-SNAP were seeded at 0.4 million cells/mL, and expression was induced with 1 µg/mL of doxycycline. At the same time, G proteins were transiently transfected and cells were incubated at 37°C for 20–24 hr. Cells were then dissociated and plated on PDL/Laminin-pre-coated glass coverslips for electrophysiological recordings. Whole-cell voltage clamp recordings were performed with the same setup, pipettes, and perfusion system as described above. The low potassium extracellular solution contained 150 mM NaCl, 5.4 mM KCl, 2 mM $CaCl_2$, 1 mM $MgCl_2$, 10 mM D-glucose, 10 mM HEPES-NaOH (pH 7.4) (~290 mOsm). The extracellular solution was exchanged to high $K^+$ solution containing 53 mM NaCl, 100 mM KCl, 1 mM $CaCl_2$, 1 mM $MgCl_2$, 10 mM D-glucose, 10 mM HEPES-NaOH (pH7.4) (~290 mOsm). The pipette solution contained 9 mM NaCl, 140 mM K-gluconate, 2 mM $MgCl_2$, 1.5 mM EGTA-K, 10 mM HEPES-KOH (pH7.4), 3 mM MgATP, 0.05 mM $Na_2GTP$ (~310 mOsm).

## Whole-cell voltage clamp recordings on Sf9 cell

The human M2R, β2AR, mouse β1AR, and human GIRK4 were cloned into a pFB vector for insect cell expression. A PreScission protease cleavage site, an enhanced green fluorescent protein (eGFP) and a deca-histidine tag were placed at the C-terminus of each construct. Sf9 cells were co-infected with P3 baculovirus with GPCRs and GIRK4 and incubated at 27°C for 40–48 hr. Whole-cell voltage clamp recordings were performed with the same system, pipettes, and perfusion system as described above. The low potassium extracellular solution contained 135 mM NaCl, 10 mM KCl, 4 mM $CaCl_2$, 5 mM $MgCl_2$, 10 mM MES-KOH (pH 6.4) (~320 mOsm). The high potassium extracellular solution contained 45 mM NaCl, 100 mM KCl, 4 mM $CaCl_2$, 5 mM $MgCl_2$, 10 mM HEPES-KOH (pH6.4) (~300 mOsm). The pipette solution contained 85 mM KCl, 60 mM KF, 1 mM $MgCl_2$, 5 mM EGTA-K, 10 mM HEPES-KOH (pH7.2), 3 mM MgATP, 0.05 mM $Na_2GTP$ (~320 mOsm).

## Whole-cell voltage clamp recordings on HEK-293T cells expressing GPCR-GIRK concatemers

Full-length human GIRK4 was fused to the C-terminus of full-length human M2R or β2AR. A serotonin 5-HT cleavable signal peptide and a Halo tag were fused to the N-terminus of each concatemer. Additionally, a SNAP tag was fused to the C-terminus of each concatemer. Concatemers were transiently transfected and cells were incubated at 37°C for 20–24 hr. Cells were then dissociated and plated on PDL/Laminin-pre-coated glass coverslips for electrophysiological recordings. Whole-cell voltage clamp recordings were performed with the same system, pipettes, perfusion system, and solutions as described above.

## Whole-cell voltage clamp recordings on HEK-293T cells expressing TRPM3 channels

Mouse TRPM3α2 was cloned into a pEG BacMam vector. A PreScission protease cleavage site, an enhanced green fluorescent protein (eGFP), and 1D4 peptide tag were placed at the C-terminus of the TRPM3 construct. TRPM3-eGFP, Sero-SNAP-GPCR, and G proteins were transiently transfected to HEK-293T cells and cells were incubated at 30°C for 48–72 hr. Cells were then dissociated and plated on PDL/Laminin-pre-coated glass coverslips for electrophysiological recordings. Whole-cell voltage clamp recordings were performed as described above. The currents were recorded using a ramp protocol from −100 mV to +100 mV, applied every second, and the currents at +100 mV were plotted. TRPM3 currents were evoked by 10 μM pregnenolone sulfate (PS) (Tocris).

## Western blot

Untransfected HEK-293T cells or HEK-293T cells transiently transfected with GPCR-GIRK4 concatemers were centrifuged and mixed with an equal volume of loading buffer containing 4% SDS and 10% β-mercaptoethanol. Samples were then run using standard SDS-PAGE procedures on Invitrogen NuPAGE 4–12% Bis-Tris gels and transferred onto PVDF membranes. Western Blot was performed using an anti-SNAP-tag (NEB), anti-HaloTag (Promega), anti-G$\alpha_{i1}$ (abcam, ab140125), anti-G$\alpha_{i2}$ (abcam, ab157204), anti-G$\alpha_o$ (Santa Cruz Biotechnology, sc-13532), or anti-G$\alpha_s$ (Santa Cruz Biotechnology, sc-383).

## cAMP quantification assay

Untransfected HEK-293T cells or HEK-293T cells transfected with βARs were cultured in 12-well plates for 20–24 hr. Sf9 cells infected with P3 baculovirus of βARs were cultured in 12-well plates for 40–48 hr. Cells were treated with either 10 μM isoprenaline or propranolol for 10 min and washed twice with PBS + 500 μM isobutylmethylxanthine (IBMX). Cells were collected in 200 μL PBS + IBMX, exposed to four freeze-thaw cycles, and centrifuged (14,000 rpm) for 10 min at 4°C. The supernatant was analyzed for cAMP content according to the manufacturer's protocol (cAMP ELISA Detection Kit, GeneScript).

## BRET sample preparation

pCEH plasmids encoding Sero-SNAP-M2R, Sero-SNAP-β2AR, Sero-SNAP-β1AR, and Sero-SNAP-D2R were used in BRET experiments. For Gα-Venus constructs, Venus was inserted to either the αa-αb loop (between residues 91 and 92 for G$\alpha_{i1}$ and 113 and 114 for G$\alpha_s$) or the αb-αc loop (between residues 121 and 122 for G$\alpha_{i1}$ and 144 and 145 for G$\alpha_s$) with flanking SGGGS linkers. Human G$\alpha_{i1}$, G$\alpha_s$, G$\alpha_{i1}$(αa-αb)-Venus, G$\alpha_{i1}$(αb-αc)-Venus, G$\alpha_s$(αa-αb)-Venus, and G$\alpha_s$(αb-αc)-Venus were cloned into a pCEH vector. Venus 156–239-G$\beta_1$ and Venus 1–155-G$\gamma_2$ were cloned into a pCEH-IRES vector to allow for expression of Gβγ-Venus from a single plasmid. Nano Luciferase-G$\beta_1$ (NLuc-G$\beta_1$) and G$\gamma_2$ were cloned into a pCEH-IRES vector to allow for expression of Gβγ-NLuc from a single plasmid. masGRK3ct, masGRK3ct-NLuc, GIRK4-NLuc, and Kir2.2-NLuc were also cloned into a pCEH vector.

For the BRET measurements between Gβγ-Venus and GIRK4-NLuc, 0.35 million HEK-293T cells were plated in each well of 12-well plates and incubated overnight at 37°C. After overnight incubation, cells were transfected with Sero-SNAP-GPCR (90 ng), Gβγ-Venus (90 ng), GIRK4-NLuc (90 ng) and different amounts of Gα (90 ng × 0, 1, 2, and 4) using Lipofectamine 2000 (ThermoFisher). Transfected cells were incubated for 20–24 hr at 37°C and then used for BRET measurements. GIRK4-NLuc was replaced by masGRK3ct-NLuc or Kir2.2-NLuc for control samples.

For BRET measurements between Gα-Venus and Gβγ-NLuc, HEK-293T cells were transfected with Sero-SNAP-GPCR (90 ng), Gβγ-NLuc (90 ng), masGRK3ct (90 ng), and Gα-Venus (90–450 ng), and incubated for 20–24 hr at 30°C or 37°C. The measured light emitted by Gβγ-NLuc is proportional to the amount of Gβγ-NLuc in the sample, and the measured light emitted by Gα-Venus is proportional to the amount of G protein trimers in the sample. By having equal intensities for Gβγ-NLuc and Gα-Venus (i.e. NLuc intensity and basal BRET ratio), the rate of Gβγ release can be compared and contrasted for different GPCRs (*Table 1*). Therefore, samples of each GPCR were prepared with different transfected Gα-Venus-DNA amounts (90–450 ng) to carry out these experiments.

## BRET measurements

After 20–24 hr incubation, transfected HEK-293T cells were washed with PBS twice and detached by incubation in PBS + 5 mM EDTA for 5 min at room temperature. Cells were harvested by centrifugation at 300 g for 3 min and resuspended into 350 µL BRET buffer (PBS supplemented with 0.5 mM $MgCl_2$ and 0.1% D-glucose). 25 µL of the suspension containing ~70,000 cells was transferred to each well in a 96-well flat-bottom white microplate (Greiner CELLSTAR). The NLuc substrate (Promega) was diluted into the BRET buffer according to the manufacturer's protocol, and 25 µL of diluted NLuc substrate were added to the cells in 96-well plates. BRET measurements were made with a microplate reader (Synergy Neo, BioTek) equipped with two emission photomultiplier tubes. The BRET signal was determined by calculating the ratio of the light emitted by Venus (535 nm with a 30 nm band width) to the light emitted by NLuc (475 nm with a 30 nm bandwidth).

## Expression and purification

Human full-length GIRK4 was cloned into a pEG BacMam vector (*Goehring et al., 2014*). A PreScission protease cleavage site, an enhanced green fluorescent protein (eGFP) and a 1D4 peptide tag were placed for purification at the C-terminus of the GIRK4 construct. For overexpression and protein purification, HEK-293S GnTl⁻ cells were grown in suspension, infected with P3 BacMam virus of the GIRK4-1D4 and incubated at 37°C. At 8–12 hr post-infection, 10 mM sodium butyrate was added to the culture, and cells were harvested 60 hr post-transduction. Cells were harvested by centrifugation, frozen in liquid $N_2$, and stored at −80°C until needed. Frozen cells were solubilized in 50 mM HEPES (pH 7.35), 150 mM KCl, 4% (w/v) n-decyl-β-D-maltopyranoside (DM), and the protease inhibitor cocktail (0.1 mg/mL pepstatin, 1 mg/mL leupeptin, 1 mg/mL aprotinin, 0.1 mg/mL soy trypsin inhibitor, 1 mM benzamidine, and 1 mM phenylmethylsulfonyl fluoride). After 2 hr of solubilization, lysed cells were centrifuged at 36,000 g for 30 min and the supernatant was incubated with 1D4 affinity resin for 1 hr at 4°C with gentle mixing. The resin was loaded onto a column and washed with buffer A (50 mM HEPES [pH 7.0], 150 mM KCl, 0.4% [w/v] DM). 5 mM DTT and 1 mM EDTA were added, and eGFP and affinity tags were cut with PreScission protease overnight at 4°C. The cleaved protein was then concentrated and run on a Superose 6 10/300 GL gel filtration column in 20 mM Tris-HCl (pH 7.5), 150 mM KCl, 0.2% (w/v) DM, 20 mM DTT, and 1 mM EDTA.

Human lipid-anchored $G\beta_1\gamma_2$, and soluble $G\beta_1\gamma_2$ were purified as described previously (*Wang et al., 2014*).

Human full-length $G\alpha_{i1}$ $G\alpha_{i2}$, $G\alpha_{i3}$, $G\alpha_o$, and $G\alpha_s$ were cloned into a pET28a vector. A PreScission protease cleavage (PPX) site followed by a deca-histidine tag was fused to the N-terminus of $G\alpha$. The $His_{10}$-PPX-$G\alpha$-pET28a vector was transformed into BL21(DE3) *E. coli* cells and transformants were cultured in LB medium containing 50 µg/L of kanamycin at 37°C for 4 hr. Isopropyl-thio-β-D-galactopyranoside was added to a final concentration of 0.5 mM to induce protein expression. Following an additional incubation at 25°C for 12 hr, the cells were harvested by centrifugation and resuspended in buffer B (200 mM HEPES-NaOH [pH 7.5], 300 mM NaCl, 2 mM $MgCl_2$, and 10 µM MgGDP) and a protease inhibitor cocktail. Cell extracts were obtained by sonication followed by centrifugation at 36,000 g for 30 min. The supernatant was incubated with Talon metal affinity resin (Clontech) for 1 hr at 4°C with gentle mixing. The resin was washed in batch with five column volumes of buffer B, then loaded onto a column and further washed with 10 column volumes of buffer B + 20 mM imidazole. The column was then eluted with buffer B + 200 mM imidazole.

For Western Blotting analysis, the eluted protein was concentrated and run on a Superdex 200 10/300 GL gel filtration column in 10 mM potassium phosphate (pH 7.4), 150 mM KCl, 2 mM $MgCl_2$, and 10 µM GDP.

For the planar lipid bilayer experiment, the eluted protein was concentrated and run on a Superdex 200 10/300 GL gel filtration column in 10 mM potassium phosphate (pH 7.4), 150 mM KCl, and 2 mM $MgCl_2$. 1 mM GTP-γS was then added to ~1 mg/mL purified proteins and incubated at 37°C for 30 min to produce $His_{10}$-PPX-$G\alpha$(GTP-γS). Residual amounts of $His_{10}$-PPX-$G\alpha$(GDP) affect the results of the subsequent bilayer experiment described below. Therefore purified $His_{10}$-PPX-$G\alpha$ (GTP-γS) was mixed with soluble $G\beta\gamma$ at a ratio of 4:1 (molar:molar) to chelate all the possibly contaminating $His_{10}$-PPX-$G\alpha$(GDP). This low concentration of $G\beta\gamma$ does not affect GIRK activity.

## Reconstitution of proteoliposomes

All lipids were purchased from Avanti Polar Lipids (Alabaster, AL). Proteoliposomes were reconstituted as described previously (*Wang et al., 2014*). In brief, 20 mg/mL of the lipid mixture (3:1 [wt: wt]=1-palmitoyl-2-oleyl-sn-glycero-3-phosphoethanolamine [POPE] : 1-palmitoyl-2-oleyl-sn-glycero-3-phospho-[1'-rac-glycerol] [POPG]) was dispersed by sonication and solubilized with 20 mM DM.

Purified GIRK4 channels and Gβγ were combined with lipid mixtures at a ratio of GIRK4:Gβγ:lipid (wt:wt:wt)=1:0.1:10. The protein-lipid mixtures were then diluted into reconstitution buffer (10 mM potassium phosphate [pH 7.4], 150 mM KCl, 1 mM EDTA, and 3 mM DTT) to make 1 mg/mL GIRK4, 0.1 mg/mL Gβγ, and 10 mg/mL (lipid mixture). Detergent was removed by dialysis against the reconstitution buffer at 4°C for 4 days.

## Planar lipid bilayer recordings

Bilayer membranes were made as previously described (*Wang et al., 2016*). In brief, 1,2-dioleoyl-sn-glycero-3-phosphoetanolamine (DOPE) and 1-palmitoyl-2-oleyl-sn-glycero-3-phosphocholine (POPC) were mixed at a 1:1 ratio (wt:wt) and doped with 3% DGS-NTA (mole fraction). A lipid solution at 20 mg/mL was then prepared using decane. This solution was painted over a ~ 120 µm hole on a piece of transparency film to form a lipid bilayer. The same recording buffer (10 mM potassium phosphate [pH 7.4], 150 mM KCl, and 2 mM $MgCl_2$) was used in both chambers. Voltage across the lipid bilayer was clamped using an Axopatch 200B amplifier (Molecular Devices, Sunnyvale, CA) in whole-cell mode. The analog current signal was low-pass filtered at 1 kHz (Bessel) and digitized at 20 kHz with a Digidata 1440A digitizer (Molecular Devices). Digitized data were recorded using the software pClamp (Molecular Devices).

After forming a lipid bilayer, proteoliposomes containing GIRK4 and Gβγ at a ratio of 1:0.1 (wt: wt) were applied to the bilayer multiple times until they fused to the bilayer. The channels were then activated by adding 32 µM C8-$PIP_2$ and 8 mM NaCl to the chamber. GIRK4 channels were partially activated in this condition (*Figure 5—figure supplement 1A.*). 300 µM GTP-γS and 2 µM soluble Gβγ were added to the chamber to chelate possible contamination of $His_{10}$-Gα(GDP). This low concentration of added Gβγ does not affect GIRK activity. 500 µM $NiSO_4$ was added directly to the bilayer twice to charge DGS-NTA lipids with $Ni^{2+}$. A solution of 30 µM $His_{10}$-Gα(GTP-γS) supplemented with 32 µM C8-$PIP_2$ was then perfused directly to the bilayer membrane several times until no further blockage was observed. Given the affinity of $His_{10}$-Gα(GTP-γS) to the bilayer containing 3% DGS-NTA lipids is ~0.5 µM, 30 µM $His_{10}$-Gα(GTP-γS) was used to saturate DGS-NTA lipids in the bilayer (*Figure 5—figure supplement 1B–1D*). The transient current decrease upon addition of $His_{10}$-Gα(GTP-γS) is due to the absence of $Na^+$ (*Figure 5C and D*).

## Confocal microscopy of HEK-293T cells

HEK-293T cells were transiently transfected with SNAP-M2R or SNAP-β2AR. After overnight incubation at 37°C, the cells were treated with 3 µM SNAP-Surface 488 (NEB) in DMEM/FBS for 30 min to stain the SNAP-tagged receptors. Cells were then washed several times with PBS, fixed with 4% paraformaldehyde, and imaged under a ZEISS inverted LSM 880 NLO laser scanning confocal microscope with an oil immersion 40 × objective (numerical aperture 1.40). Microscope and software settings were kept the same for all images acquired. The fluorophore was excited with a white light laser of 488 nm.

## Confocal microscopy of giant unilamellar vesicles (GUVs)

DOPE:POPC 1:1 (wt:wt) lipid mixture with 3% DGS-NTA lipids was used to produce GUVs. GUVs were prepared according to a published protocol (*Martinac et al., 2010*). In short, the lipid mixture was dried under a stream of argon. 2 µL of water was added to dried lipids to hydrate the lipids. After 3 min, 1 mL of 0.4 M sucrose solution was added. The lipid solution was then moved to a water bath and incubated at 42°C for 3 hr to form GUVs. To monitor the interaction between GUVs and $His_{10}$-Gα(GTP-γS), Alexa Fluor 488 labeled $His_{10}$-Gα(GTP-γS) was prepared in a buffer containing 10 mM potassium phosphate pH 7.4, 150 mM KCl, 2 mM $MgCl_2$, and 2 nM $NiSO_4$. This protein solution was then mixed with 1/50 vol of GUVs. The equator plane of GUVs was imaged using a ZEISS inverted LSM 880 NLO laser scanning confocal microscope with an oil immersion 100 × objective (numerical aperture 1.40). Microscope and software settings were kept the same for all images

acquired. The fluorophore was excited with a white light laser of 488 nm. The fluorescence intensity at the edge of the GUVs was measured using the Zeiss ZEN two software.

## Kinetic simulation

Mass balance equations were derived based on the model (*Figure 7A*). Rate constants used in the simulation are presented and referenced in *Table 2*. The set of first order differential equations was solved using the NDSolve function in Mathematica (Wolfram).

## Diffusion model

The graphs in *Figure 7D and F* (dashed) were generated by solving for the concentration of Gβγ ($C(r)$) analytically, using DSolve in Mathematica, the equation $\nabla \cdot D\nabla C(r) - kC(r) = 0$ in polar coordinates with the near boundary condition set at the perimeter of an 0.01 µm radius circle corresponding to a flux of 15.9 Gβγ µm$^{-1}$ sec$^{-1}$ (which corresponds to a single GPCR inside the circle with a turnover rate of 1 Gβγ sec$^{-1}$) and far boundary condition zero. $D$ is the diffusion coefficient (0.2 µm$^2$ sec$^{-1}$) and $k$ is the Gβγ decay constant (1 sec$^{-1}$). To model a GPCR density of 5 µm$^{-2}$, on average the small circle with a GPCR resides within a circle of radius ~ 0.25 µm. At 0.25 µm the Gβγ concentration has not decayed to zero and therefore neighboring GPCRs increase the Gβγ concentration slightly above what is shown for a lone GPCR in an essentially infinite membrane, but not enough to activate GIRK. The graphs in *Figure 7E and F* (solid curve) were generated numerically, using NDSolve in Mathematica, the equation $\nabla \cdot D\nabla C(r) + s(hotspot) - kC(r) = 0$, applying a circular finite element mesh with a zero concentration boundary condition at the perimeter, far from a smaller, central 'hotspot' circle. $s$ refers to Gβγ generation (molecules µm$^{-2}$ sec$^{-1}$) and is applied over the hotspot. The magnitude of $s$ was selected to be near the steady state value of $\frac{dc}{dt}$ in the kinetic equations. At a radius 3.16 times the hotspot radius (i.e. corresponding to a circular area 10 times the hotspot area) the Gβγ concentration is not zero (*Figure 7F*) and therefore in a cell with hotspots covering 10% of the membrane the Gβγ concentration will be slightly higher than shown for the case of a lone hotspot. *Figure 7G* graphs the $C(r = 0)$ solution to equation $\nabla \cdot D\nabla C(r) + s(hotspot) - kC(r) = 0$, solved as described.

## Acknowledgements

We thank Catherine Proenza for sharing the unpublished murine SAN isolation protocol, and David C.Gadsby for advice on SAN isolation. We thank Andrew Siliciano for advice on BRET experiments and sharing BRET constructs, Stephan Philipp for sharing mTPRM3 constructs, Thomas P. Sakmar for sharing human Gα$_{i2}$ constructs, and staff of RU resource centers (the Bio-Imaging Resource Center, the High-Throughput Screening Resource Center, and the Comparative Bioscience Center). We thank David CGadsby and members of the MacKinnon and Chen laboratory, especially Weiwei Wang for helpful discussions and Christoph Haselwandter for advice on the finite element method. We thank Chia-Hsueh Lee, Ji Sun, Michael Oldham, Emily Brown, and James Chen for advice on the manuscript.

This work was supported in part by GM43949. RM is an investigator in the Howard Hughes Medical Institute.

## Additional information

### Funding

| Funder | Grant reference number | Author |
| --- | --- | --- |
| National Institutes of Health | GM43949 | Roderick MacKinnon |

The funders had no role in study design, data collection and interpretation, or the decision to submit the work for publication.

## Author contributions
Kouki K Touhara, Conceptualization, Data curation, Software, Formal analysis, Validation, Investigation, Visualization, Methodology, Writing—original draft, Project administration, Writing—review and editing; Roderick MacKinnon, Conceptualization, Software, Formal analysis, Supervision, Funding acquisition, Validation, Visualization, Methodology, Writing—original draft, Project administration, Writing—review and editing

## Author ORCIDs
Kouki K Touhara http://orcid.org/0000-0003-3167-9784
Roderick MacKinnon http://orcid.org/0000-0001-7605-4679

## Ethics
Animal experimentation: C57BL/6J (Jackson Labs) male and female adult mice (at least 10 weeks old) were used. Animals were kept in cages with a 12:12 h light/dark cycle and unrestricted access to food and water. All experimental procedures were carried out according to a protocol approved by the Institutional Animal Care and Use Committee (IACUC) of The Rockefeller University (Protocol #16864).

## Decision letter and Author response
Decision letter https://doi.org/10.7554/eLife.42908.020
Author response https://doi.org/10.7554/eLife.42908.021

# Additional files
## Supplementary files
• Transparent reporting form
DOI: https://doi.org/10.7554/eLife.42908.018

## Data availability
All data generated or analyzed during this study are included in the manuscript and supporting files.

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
