## [Decision Letter]

Thank you for submitting your article "Molecular basis of signaling specificity between GIRK channels and GPCRs" for consideration by *eLife*. Your article has been reviewed by three peer reviewers, including Kenton Jon Swartz as the Reviewing Editor and Reviewer #1, and the evaluation has been overseen by Richard Aldrich as the Senior Editor.

The reviewers have discussed the reviews with one another and the Reviewing Editor has drafted this decision to help you prepare a revised submission.

Summary:

This is a fascinating manuscript exploring why Gα_i_ coupled GPCRs like M2R activate GIRK channels while Gα_s_coupled receptors like β2AR do not, even though both liberate the same Gβγ subunits that ultimately bind to and activate the GIRK channel. This is a systematic study by all measures, including testing a wide range of conditions, expression systems, approaches (both experimental and computational) to look at each step in the pathway to arrive at the simple and elegant solutions that the receptors that can activate GIRK channels must liberate Gβγ at faster rates and therefore achieve higher concentrations of the active subunits. The authors were also able to push the Gα_s_system to observe GIRK activation under non-physiological experimental conditions, which we found particularly satisfying. The additional controls with TRPM3 are beyond what we would have expected from a study such as this and they recapitulate the core conclusion. The modeling adds considerably by illustrating the conditions required for selective activation of GIRK channels. The authors are to be commended for putting together an unusually complete and conclusive story on a topic of fundamental biological importance. The presentation also fully takes advantage of the *eLife* format to concisely yet thoroughly present a large body of work.

Suggested revisions:

1) One point of comment is that conformational changes in both G protein and receptor proteins are required for their stable association (the *k_12_* step in Figure 7A). This seems to imply that one of these conformational changes may be the rate-limiting component of the *k_12_* receptor/G protein association step. In the second paragraph of the Discussion, data are discussed supporting this idea, identifying conformational change in the G protein, rather than receptor, as the limiting factor for receptor/G protein association. In our view this is quite a remarkable result, and it could be more strongly emphasized and discussed given its implications. For example, if conformational change in the Gα protein is the true rate-limiting step, this implies that selectivity of GIRK activation is independent of receptor expression levels, since active-state G protein subunits and not activated receptor molecules are limiting. It also means that GIRK activation should still not occur even if several Gα_s_-coupled receptors are activated simultaneously. This could be a useful biological feature of the system given the rich collection of GPCRs expressed in the heart and the nervous system.

2) Our only criticism would be that the manuscript contains a somewhat cursory treatment of the literature, and a corresponding lack of discussion of potentially alternative theories and modifying aspects. For example:

a) G-protein subunit specificities were a big story in the late 1980's and the 1990's. Some functional experiments showed remarkable subtype specificities, while biochemical experiments showed very little (with the exception of the retinal β_1_γ_1_ pair). This is not at all discussed and (unless we overlooked this) the authors don't even tell us which Gβγ they are using.

b) Paul Insel's lab did a thorough determination of the stoichiometries of the receptor/G-protein/cyclase signaling chains in the heart. Although, to our knowledge, they did not attempt an analysis in sinoatrial node cells, they did report an excess of Gα_i_ over Gα_s_ that may explain the authors' findings similarly well. Furthermore, several labs, notably Al Gilman's, determined quantities of various G proteins in various tissues, and usually found very little Gα_s_.

c) FRET and BRET experiments done in the mid-2000s have yielded controversial results whether Gα and Gβγ dissociate or just re-arrange during activation. Particularly for Gα_i_ variants, several labs reported that they do not (or do not need to) dissociate in intact cells (which appears be different in cell membrane experiments). How would such a finding affect the authors’ model and conclusions?

3) A final remark concerns the choice of reaction parameters in Table 2, which are all taken from the indicated references. However, the GTP hydrolysis rate (Breitwieser and Szabo, 1988) is very high compared with turnover numbers reported from biochemical experiments by many labs (which are on the order of 2-4/min).

4) The authors do a nice job of documenting population responses for most experiments, except for the BRET responses. Would there be a simple way to analyze the results from the most important experiments to provide population data?

5) In the Results, the authors might specifically mention the cAMP ELISA used to measure the concentration of the nucleotide simply to orient the reader without going to the methods.

6) Subsection “Direct measure of the Gβγ-GIRK interaction”, last paragraph, suggest modifying "no change" to "only very small changes".

---

## [Author Response]

Suggested revisions:1) One point of comment is that conformational changes in both G protein and receptor proteins are required for their stable association (the k_12_ step in Figure 7A). This seems to imply that one of these conformational changes may be the rate-limiting component of the k_12_ receptor/G protein association step. In the second paragraph of the Discussion, data are discussed supporting this idea, identifying conformational change in the G protein, rather than receptor, as the limiting factor for receptor/G protein association. In our view this is quite a remarkable result, and it could be more strongly emphasized and discussed given its implications. For example, if conformational change in the Gα protein is the true rate-limiting step, this implies that selectivity of GIRK activation is independent of receptor expression levels, since active-state G protein subunits and not activated receptor molecules are limiting. It also means that GIRK activation should still not occur even if several Gαs -coupled receptors are activated simultaneously. This could be a useful biological feature of the system given the rich collection of GPCRs expressed in the heart and the nervous system.

The fundamental conclusion here is that the second order rate constant *k_12_* is intrinsically faster for M2R than for the βARs and this accounts for specificity. We do not know whether this intrinsic difference is related to differences in conformational changes in the G protein trimer or the GPCR or both. However, in light of our findings, the observations of Leaney et al. suggest that rate differences are mainly ‘encoded’ by the Gα subunits. We put this idea forth only as a suggestion. This is definitely an area worth further investigation.

To the dependence of GIRK activation on receptor expression levels, our data support the conclusion that under physiological G protein concentrations, when receptors are activated by ligand they are generally not saturated with G protein trimer, especially in the case of the βARs owing to a smaller *k_12_*. This can be appreciated from the BRET data in Figure 3D and 3E and in a new figure supplement (Figure 3—figure supplement 1E), which provides the requested statistical information from multiple experiments (see below). Increasing the GPCR concentration will still increase the rate of Gβγ generation (because the forward rate of formation of activated receptor is *k_12_* [G protein trimer] [GPCR]), but a very high density of βARs would be required.

2) Our only criticism would be that the manuscript contains a somewhat cursory treatment of the literature, and a corresponding lack of discussion of potentially alternative theories and modifying aspects. For example:a) G-protein subunit specificities were a big story in the late 1980's and the 1990's. Some functional experiments showed remarkable subtype specificities, while biochemical experiments showed very little (with the exception of the retinal β_1_γ_1_ pair). This is not at all discussed and (unless we overlooked this) the authors don't even tell us which Gβγ they are using.

We present in the Materials and methods that we used human Gβ_1_ and Gγ_2_ (subsection “Whole-cell voltage clamp recordings on HEK293T and CHO cells expressing GIRK channel”, subsection “BRET sample preparation” and subsection “Expression and purification”, second paragraph). In Figure 1 and in Figure 1—figure supplement 1 we show that GPCR-GIRK signaling specificity persists in cells derived from species ranging from mammals to insects. Because G protein conservation between insects and mammals is very low compared to conservation between mammalian subtypes, it is extremely unlikely that Gβγ subtype affects signaling. We now include this point in the discussion on Gβγ subtype specificities (subsection “Gβγ specificity in native and heterologously expressed GIRK channels”, last paragraph).

b) Paul Insel's lab did a thorough determination of the stoichiometries of the receptor/G-protein/cyclase signaling chains in the heart. Although, to our knowledge, they did not attempt an analysis in sinoatrial node cells, they did report an excess of Gαi over Gαs that may explain the authors' findings similarly well. Furthermore, several labs, notably Al Gilman's, determined quantities of various G proteins in various tissues, and usually found very little Gαs.

In this study we have demonstrated that expression levels of Gα_i_ and Gα_s_ are similar in HEK cells (Figure 3—figure supplement 2), where we still observe signaling specificity (Figure 1D). Thus, given the mechanism of specificity based on an intrinsically larger *k_12_* for the Gα_i_-coupled GPCR, higher levels of Gα_i_ (compared to Gα_s_) are not required for specificity. If the concentration of Gα_i_ were to be greater than Gα_s_ in certain cell types then this would also contribute to specificity, but it is not a requirement. This is such an important point raised by the reviewers that we have added a paragraph to the Discussion (second paragraph) to address it.

c) FRET and BRET experiments done in the mid-2000s have yielded controversial results whether Gαand Gβγ dissociate or just re-arrange during activation. Particularly for Gαivariants, several labs reported that they do not (or do not need to) dissociate in intact cells (which appears be different in cell membrane experiments). How would such a finding affect the authors' model and conclusions?

As pointed out, a FRET study by Bünemann et al. suggested that Gα_i_ and Gβγ undergo subunit rearrangement rather than dissociation during G protein activation (Bünemann et al., 2003). However, in our BRET experiments we observed robust apparent dissociation of Gα and Gβγ using two different Gα-Venus constructs (Figure 6). Furthermore, and perhaps more importantly, our previous studies using reconstitution clearly demonstrate that free Gβγ activates GIRK and that Gα_i_(GDP) inhibits the channel by removing Gβγ from GIRK through equilibrium-mass action (Wang et al., 2014). Furthermore, our atomic structural studies also demonstrate that the GIRK binding site on Gβγ overlaps with the Gα binding site (Whorton and MacKinnon, 2013). Given these data, the necessity of G protein subunit dissociation seems an inescapable fact.

3) A final remark concerns the choice of reaction parameters in Table 2, which are all taken from the indicated references. However, the GTP hydrolysis rate (Breitwieser and Szabo, 1988) is very high compared with turnover numbers reported from biochemical experiments by many labs (which are on the order of 2-4/min).

Intrinsic GTPase activity of Gα is known to be 2-4 min^-1^. However, under physiological conditions GTPase-activating proteins (GAPs) accelerate the rate by ~10-1000 fold (Sprang, 2016). Breitwieser and Szabo measured the GTP hydrolysis rates in isolated atrial myocytes where endogenous GAPs were most likely present, as in our experiments.

4) The authors do a nice job of documenting population responses for most experiments, except for the BRET responses. Would there be a simple way to analyze the results from the most important experiments to provide population data?

We added Figure 3—figure supplement 1E to provide population data for BRET experiments in Figure 3.

5) In the Results, the authors might specifically mention the cAMP ELISA used to measure the concentration of the nucleotide simply to orient the reader without going to the methods.

We added the information to the manuscript (subsection “Gβγ specificity in native and heterologously expressed GIRK channels”, last paragraph).

6) Subsection “Direct measure of the Gβγ-GIRK interaction”, last paragraph, suggest modifying "no change" to "only very small changes".

We made this change.